# Constrained patterning of orientated metal chalcogenide nanowires and their growth mechanism

Qishuo Yang [1,2,3], Yun-Peng Wang[4], Xiao-Lei Shi[5], XingXing Li[1], Erding Zhao[1], Zhi-Gang Chen [5], Jin Zou [6], Kai Leng [7], Yongqing Cai [8], Liang Zhu[1] ✉, Sokrates T. Pantelides [9,10] & Junhao Lin [1,2] ✉

One-dimensional metallic transition-metal chalcogenide nanowires (TMC-NWs) hold promise for interconnecting devices built on two-dimensional (2D) transition-metal dichalcogenides, but only isotropic growth has so far been demonstrated. Here we show the direct patterning of highly oriented $Mo_6Te_6$ NWs in 2D molybdenum ditelluride ($MoTe_2$) using graphite as confined encapsulation layers under external stimuli. The atomic structural transition is studied through in-situ electrical biasing the fabricated heterostructure in a scanning transmission electron microscope. Atomic resolution high-angle annular dark-field STEM images reveal that the conversion of $Mo_6Te_6$ NWs from $MoTe_2$ occurs only along specific directions. Combined with first-principles calculations, we attribute the oriented growth to the local Joule-heating induced by electrical bias near the interface of the graphite-$MoTe_2$ heterostructure and the confinement effect generated by graphite. Using the same strategy, we fabricate oriented NWs confined in graphite as lateral contact electrodes in the 2H-$MoTe_2$ FET, achieving a low Schottky barrier of 11.5 meV, and low contact resistance of 43.7 $\Omega\,\mu m$ at the metal-NW interface. Our work introduces possible approaches to fabricate oriented NWs for interconnections in flexible 2D nanoelectronics through direct metal phase patterning.

Contact issues have been a critical bottleneck impeding the performance of devices based on two-dimensional (2D) transition-metal dichalcogenides (TMDs)[1,2]. Upon the formation of metal-semiconductor contacts at the interface between electrodes and the material, Schottky barriers form due to differences in work functions or the presence of metal-induced gap states, which lead to deteriorated contact

properties[3-6]. Aiming to enhance the device performance of semiconducting TMDs, previous studies have proposed strategies to lower the Schottky barriers, such as fabricating semimetal layers like Bi between metals and semiconducting TMDs[7] or fabricating metallic phases as contacts to the TMDs devices[8,9], both of which have demonstrated effective alleviation of the contact problem. However, contact

[1]Department of Physics and Shenzhen Key Laboratory of Advanced Quantum Functional Materials and Devices, Southern University of Science and Technology, Shenzhen, People's Republic of China. [2]Quantum Science Center of Guangdong-Hong Kong-Macao Greater Bay Area (Guangdong), Shenzhen, People's Republic of China. [3]School of Mechanical and Mining Engineering, The University of Queensland Brisbane, Qld, Australia. [4]School of Physics and Electronics, Hunan Key Laboratory for Super-Micro Structure and Ultrafast Process, Central South University, Changsha, People's Republic of China. [5]School of Chemistry and Physics, Queensland University of Technology Brisbane, Qld, Australia. [6]Center for Microscopy and Microanalysis, The University of Queensland Brisbane, St Lucia, Qld, Australia. [7]Department of Applied Physics, Hong Kong Polytechnic University, Hung Hom, Kowloon, Hong Kong, China. [8]Institute of Applied Physics and Materials Engineering, University of Macau, Taipa, Macau, SAR, China. [9]Department of Physics and Astronomy, Vanderbilt University, Nashville, TN, USA. [10]Department of Electrical and Computer Engineering, Vanderbilt University, Nashville, TN, USA. ✉e-mail: zhul6@sustech.edu.cn; linjh@sustech.edu.cn

properties are affected not only by the Schottky barriers, but also by factors like contact area, interfacial cleanliness, chemical affinity, etc., thus, still posing challenges in further enhancing device performance.

As a type of material that can be directly obtained on TMDs without the need for additional metal deposition, metallic transition-metal chalcogenide nanowires (TMC-NWs) with a general formula of $X_6Y_6$ ($X$ = Mo, W; $Y$ = S, Se, and Te)[10–12] have been considered as potential candidates for improving device contacts. These NWs exhibit tunable metallic characteristics due to the strong hybridization between $d$- and $p$-orbitals of the $X$ and $Y$ elements[10,12–15], while at the same time, demonstrating great structural stability and flexibility under extreme conditions[14]. Additionally, previous reports have shown that assembling TMC-NWs with TMDs in field-effect transistors (FETs) effectively lowers the Schottky barrier height between the TMDs and the metal electrode[16]. These unique features of TMC-NWs position them as promising conductive interconnects for future nanodevices.

The fabrication of TMC-NWs can be achieved in various means, in 2014, Lin et al. reported the fabrication of isolated single NW through nano-modification by in-situ electron beam to the $MoS_2$, $MoSe_2$, and $Mo_xW_{1-x}S_2$ in an aberration corrected scanning transmission electron microscope (STEM)[14,17]. In addition, direct chemical-synthesis methods including physical vapor deposition (PVD)[15], molecular beam epitaxy (MBE)[18], metal-organic chemical vapor deposition (MOCVD)[16], and atmospheric pressure chemical vapor deposition (APCVD)[19] have been used to prepare relatively large-scale $Mo_6X_6$ NW bundles, nanoplates, or nanonetworks. More recently, Hong et al.[20] and Zhu et al.[21]. showed that a transition from TMDs to NWs occurs at high temperatures by in-situ heating or high-temperature annealing to induce chalcogen vacancies in 2D TMDs[22]. Although significant progress has been made in the growth of TMC-NWs, their orientation control is still challenging due to its 1D nature with unrestricted unidirectional growth mode, resulting in an isotropic distribution of NW bundles. A key challenge towards its practical use in devices is to achieve direct patterning of oriented NWs arrays, which can serve as lateral metallic contacts between the metal electrode and the device. Nevertheless, growth of TMC-NWs with defined orientation remains elusive.

In this work, we report the direct patterning of highly oriented $Mo_6Te_6$ NWs transitioning from 2D molybdenum ditelluride ($MoTe_2$) using graphite as confined encapsulation layers under external stimuli. To study the growth mechanism of the oriented NWs, we apply in-situ electrical bias through the confined graphite layers to $MoTe_2$ to induce structural transition in a STEM. Atom-resolved STEM imaging confirms that the directional growth of $Mo_6Te_6$ NW bundles starts at the edge of the graphite-sandwiched 2D $MoTe_2$ and keep growing along a specific facet without deflecting. Combining with series of controlled in-situ experiments and first-principles calculations, we confirm that the growth of highly oriented NWs originates from both the confinement effect of the graphite electrodes and the bias-voltage-induced local Joule-heating. The Joule heating generates Te vacancies in $MoTe_2$ and further induces structural transition from 2H $MoTe_2$ to $Mo_6Te_6$ NWs, while the ejected Te atoms are forced by the confining graphite sheets to migrate anisotropically along the specific confined channels without accumulation. Transport results of fabricated FET devices using the oriented NWs as lateral contact electrodes demonstrate a significant improvement in the contact properties compared with previous works, reaching a low Schottky barrier of 11.5 meV, and contact resistance of 43.7 $\Omega\,\mu$m at the metal-NW interface. This work offers novel approaches to fabricate highly oriented NWs through introducing confinement effect and provide in-depth understanding of the structural evolution of 2D $MoTe_2$ under external bias. The enhanced contact properties confirmed in the transport results also demonstrate their potential applications in nanoelectronics.

## Results

### Confinement strategy for oriented growth of $Mo_6Te_6$ NWs

Thermal annealing of TMDs is widely used in fabricating TMC-NWs in the reported synthesis strategies[17,18,21]. When annealing is applied to the unconfined $MoTe_2$ flakes, Te vacancies are generated throughout the flake, leading to the random in-plane nucleation of NWs. With continuous heating, NWs grow freely along different zigzag directions of the original lattice, which leads to the formation of a disordered NW network as demonstrated in the schematic of Fig. 1a, c.

To regulate the growth orientation of $Mo_6Te_6$ NWs to achieve oriented patterning, random in-plane nucleation must be restrained. Encapsulation experiments[23] and theory[24] show significant energy barriers for traversing the graphene hexagonal lattice even by small atoms like He and H. Inspired by this, we propose that confining the $MoTe_2$ flake with graphite layers may restrict the generation of Te vacancies and also their migration pathways. Thus, the confining strategy may change the random in-plane nucleation of NWs to confined nucleation at the edge and would achieve the regulation of the growth orientation from isotropic to along certain specific orientations, as indicated in the schematic of Fig. 1d, f.

### In-situ electrical-bias-induced oriented NWs growth in $MoTe_2$

Phase transitions can be realized by introducing various external stimulations in TMDs[8,25,26]. To verify the above-proposed strategy, we apply in-situ electrical bias to a freestanding graphite-sandwiched 2D $MoTe_2$ heterostructure and simultaneously record its structural evolution in a STEM. Figure 2a presents the schematic diagram of the heterostructure, fabricated on a commercial microelectromechanical-system (MEMS) chip with the corresponding current flow when an electrical bias is applied. A detailed optical-microscope image of such a heterostructure is shown in Suppl. Figure 1. The heterostructure is fabricated using a polycarbonate (PC) microdome transfer method conducted by a 2D transfer platform (Suppl. Figure 2), while the few-layer graphite and $MoTe_2$ flakes are prepared by a standard scotch-tape cleavage method (see "Methods"). Since $MoTe_2$ is sensitive to oxygen, all sample fabrications are conducted in a nitrogen-protected glove box to prevent contamination and oxidation. The top and bottom graphite layers in the heterostructure are connected to the electrode that is prefabricated on the MEMS chip, which generates a uniform electric field in the heterostructure region where $MoTe_2$ has graphite on both sides. Moreover, the top and bottom graphite electrodes are sufficiently thin so that they do not disrupt the atomic imaging of the $MoTe_2$ structural evolution.

Figure 2b, d shows a series of low-magnification high-angle annular dark-field (HAADF)-STEM images taken at different stages of the oriented growth of $Mo_6Te_6$ NWs. Figure 2b is acquired before applying the bias voltage. We confirm that the sample is 2H-$MoTe_2$ by the HAADF image (Suppl. Figure 3). By slowly increasing the applied bias voltage to 2.5 V, a stripe-like 1D structure is suddenly observed at the sandwiched heterostructure edge and keeps extending into the middle region of the hole when maintaining the bias voltage. Although slight deflection of certain stripes is observed due to surface undulations (see Suppl. Figure 4), most of the stripes' growth direction remains unchanged (see Fig. 2c). By further increasing the applied bias voltage to 4 V, the growth of the new structure becomes faster along the same growth direction (see Fig. 2d). Additional selected area electron diffraction (SAED) and Raman spectra confirm the conversion of 2H $MoTe_2$ into $Mo_6Te_6$ NWs, as shown in Suppl. Figure 5, 6.

Figure 2e shows the corresponding time-resistance curve record by the MEMS chip during the STEM imaging process in Fig. 2b, d. As can be seen, when the bias voltage reaches 2.5 V, a relatively large drop in resistance is observed. This indicates that a structural-transformation-induced resistance drop occurred in our experiment. It should be noted that below 2.5 V, when bias voltage is stable, the

resistance remains constant (see Suppl. Fig. 7), suggesting that an activation threshold of voltage (in our case 2.5 V) is required for the formation of NWs.

To determine the structural changes of $MoTe_2$, we use a focused ion beam (FIB) to make a cross-sectional sample near the as-grown structure in the region marked by a red rectangle in Fig. 2d. The thicknesses of the top and bottom graphite layers are 5.7 and 5.9 nm, respectively, while the $MoTe_2$ film is 15.8 nm (Suppl. Fig. 8). Figures 2f, g shows cross-sectional HAADF images of the original 2H-$MoTe_2$ and the converted $Mo_6Te_6$ NW bundles. The conversion of a 2D flake to 1D bundle does not generate much volume change, mainly due to the confinement effect between the graphite electrodes. Zoom-in images of the interface between graphite/$MoTe_2$ and graphite/$Mo_6Te_6$ NW are shown in Fig. 2h, i with the atomic-lattice model overlaid, respectively. $MoTe_2$ is in perfect 2H stacking in which the orientations of the prismatic unit cells are inverted between layers, while the converted NWs also maintain a close-packed bundle structure as reported in ref.[21]. Moreover, the interface between the confined graphite layers and $MoTe_2$ is atomically flat (Fig. 2h), demonstrating the high cleanliness of our transfer method, as no contamination is introduced into the interface during the sample preparation process. It is assumed that the flat interface generates a uniform vertical electric field when an in-situ electric bias is applied. Such a flat interface is preserved during the conversion, indicating a rapid reaction process. Meanwhile, because of the different unit cell parameters of 2H-$MoTe_2$ and $Mo_6Te_6$ NW (c axis of $Mo_6Te_6$ NWs is around 7.6 Å and 2H-$MoTe_2$ is 7 Å)[21], mismatching is present between 2D layers and 1D NWs at the lateral connection (Suppl. Fig. 9). The oriented growth behaviour of NWs remains consistent across varying thicknesses of graphite and $MoTe_2$ layers, as long as the $MoTe_2$ is graphite-confined on both sides (see Suppl. Fig. 8b for another device with different thickness of graphite layers).

Because bulk $Mo_6Te_6$ NWs are metallic[21], the appearance of NWs in $MoTe_2$ reduces the resistance of the device, which explains the sudden resistance drop in Fig. 2e. Furthermore, the X-ray energy-dispersive spectroscopy (EDS) results also confirm that the Mo:Te composition of the stripe-like structure is about 1:1 (Suppl. Fig. 10), indicating the complete conversion from 2D $MoTe_2$ to 1D $Mo_6Te_6$ NWs.

### In-situ STEM study of the oriented growth of $Mo_6Te_6$ NWs

To understand the structural conversion from 2H-$MoTe_2$ to $Mo_6Te_6$ NWs, we investigate the dynamic growth of $Mo_6Te_6$ NWs by in-situ STEM characterizations. Figures 3a, b shows the optical microscope image and the zoom-in HAADF-STEM image of a free-standing hole region of the graphite-sandwiched $MoTe_2$. In Fig. 3b, the red line represents the edge of the top-layer graphite flake (depicted in Fig. 3c). Note that in the region where $MoTe_2$ is covered by graphite on both sides (above red line), oriented bundle-like $Mo_6Te_6$ NWs are converted from $MoTe_2$ under bias modulation. However, the lack of double-sided graphite coverage of $MoTe_2$ (below the red line) result in an amorphous film composed of random size Mo-rich clusters, as shown schematically in Fig. 3d (structure and EDS results are shown in Suppl. Fig. 11).

During the growth of oriented NWs, a series of high-resolution STEM images are taken at the growth frontier of the NWs bundles as shown in Fig. 3e, which clearly indicates that the growth direction of $Mo_6Te_6$ NWs is along specific zigzag edge ((100) facet) of 2H-$MoTe_2$. Figure 3e also reveals that the NWs remain terminated at the (110) facet of the 2H-$MoTe_2$ during the growth. Moreover, the interface is quite sharp without Moiré overlapping patterns, indicating a rapid and complete phase transition under the electrical biasing.

To unveil the dynamical behaviours of NWs at a larger scale, Fig. 3f shows a series of low-magnification STEM images. The

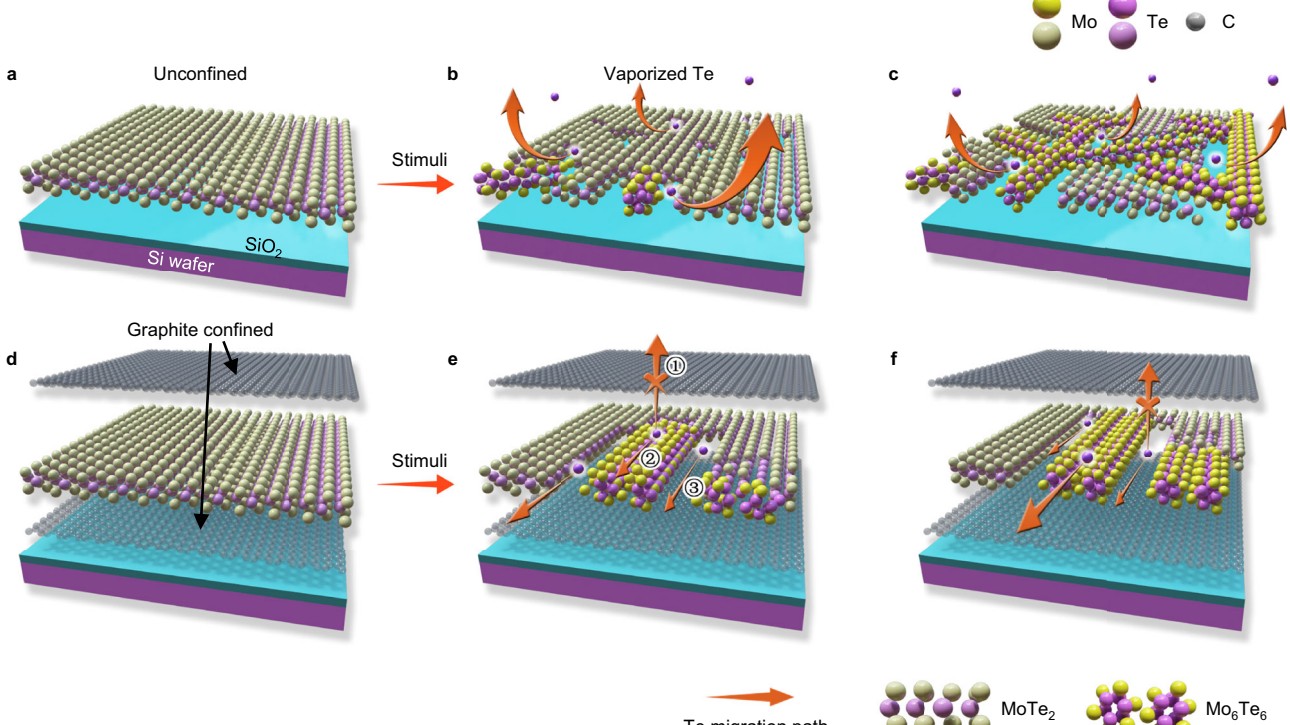

**Fig. 1 | Control growth of oriented $Mo_6Te_6$ NWs through introducing graphite confined layers. a–c** Schematics of the conversion of disordered NWs through applying thermal annealing. **d–f** Schematics of the conversion of oriented NWs in the graphite-encapsulated 2H-$MoTe_2$.

Potential Te migration pathways in a graphite-confined $MoTe_2$ flake are highlighted by orange arrows in (**e**) including 1) penetrating graphite layers, 2) along NWs, and 3) along the gaps between NW bundles.

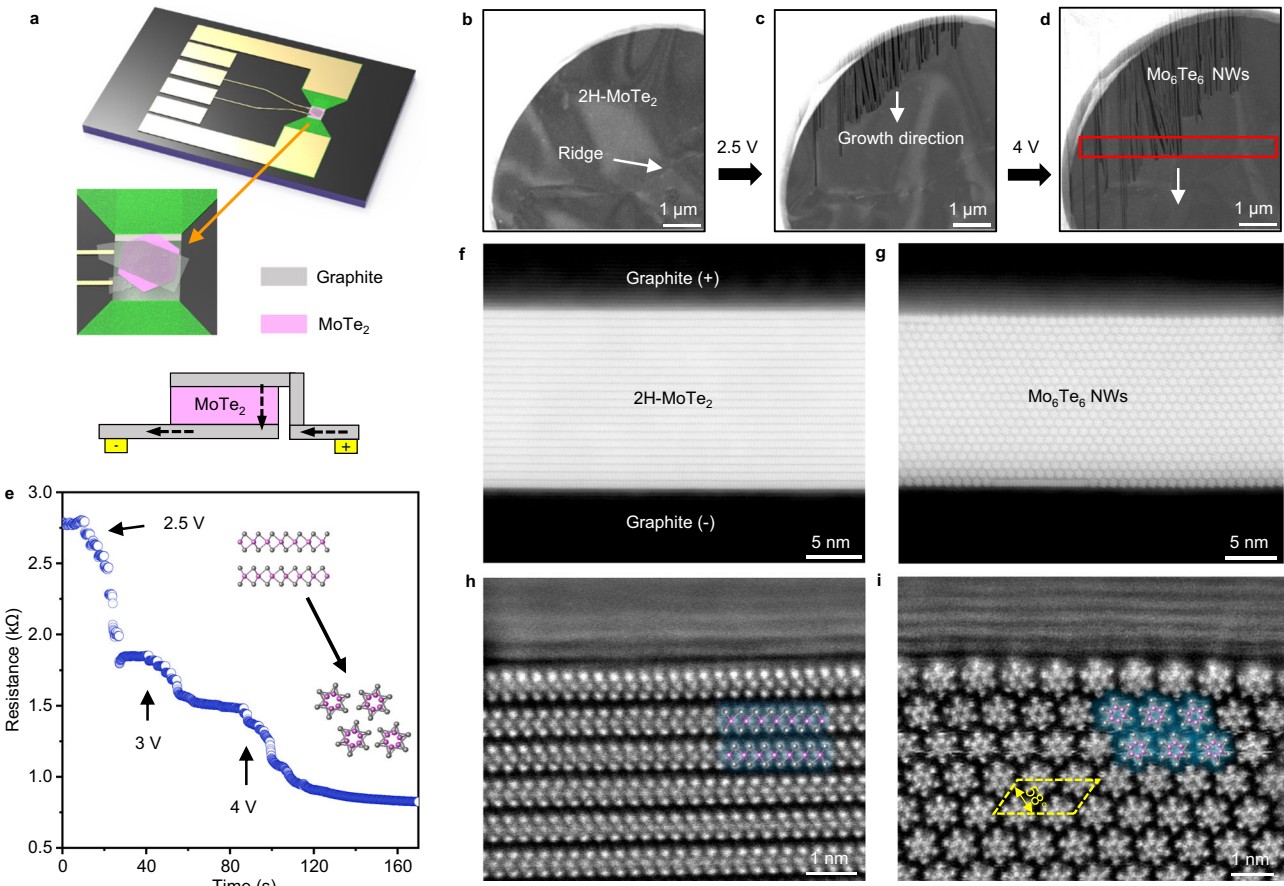

**Fig. 2 | In-situ growth of oriented $Mo_6Te_6$ NW bundles in 2H-$MoTe_2$ confined by graphite electrodes under electrical biasing. a** Schematic depiction of the graphite-$MoTe_2$-graphite heterostructure device on the in-situ microelectromechanical system (MEMS) chip. The graphite layers on the top and bottom are connected to the positive and negative electrodes of the chip, respectively. The current flow is highlighted by black dashed arrows, and the electrodes on the MEMS chip is highlighted by yellow. **b**–**d** A series of high-angle annular dark-field (HAADF) scanning transmission electron microscopy (STEM) images showing the dynamic growth of the oriented NW bundles. White arrow indicates surface undulations as reference marker during structure conversion. **e** The evolution of resistance as a function of time measured through the chip during the in-situ experiment with the schematic indicating the phase transition inserted. A rapid drop in resistance and the growth of NWs were observed simultaneously when the voltage reached 2.5 V. **f**, **g** Cross-sectional HAADF images of the 2H-$MoTe_2$ (**f**) and the as-grown $Mo_6Te_6$ NW bundle (**g**). **h**, **i** Zoom-in atomic images showing the clean interface of 2H-$MoTe_2$/graphite (**h**) and $Mo_6Te_6$/graphite (**i**) with the atomic models overlaid. Source data are provided as a Source Data file.

lateral connection between $Mo_6Te_6$ NW and $MoTe_2$ is steplike. We observe in Fig. 3e that the $MoTe_2$ (110) facet, where NWs are terminated, is continuously transformed into $Mo_6Te_6$ NWs, leaving the (100) facet exposed and elongated. In other words, the growth of the NWs does not deflect due to such special growth behaviour. Instead, as the NW grow, $MoTe_2$ is selectively consumed at the (110) facet so that the NWs can elongate only along the edge of the $MoTe_2$ (100) facet. Note that the conversion is not 2D-to-2D as gaps with random widths can be observed between NW bundles. The formation of these gaps is because almost half of the Te atoms in $MoTe_2$ are ejected during the structural transformation from $MoTe_2$, which results in a huge volume reduction relative to the initial state.

**Graphite confinement effect and growth mechanism**

To interrogate the confinement effect of the graphite layers step by step, we first conduct controlled in-situ heating experiments. The thickness of the top and bottom graphite layers is around 15 and 20 nm for $MoTe_2$. The results are shown in Fig. 4a, b and Suppl. Figs. 12, 14. We find that global heating also converts the double-sided graphite-confined 2D $MoTe_2$ flake into oriented NWs at 600 °C. However, incomplete conversion where $MoTe_2$ and NWs have substantial overlaps is observed at the contact region (pointed by yellow

arrows in Fig. 4a; more information is shown in Suppl. Fig. 13), while the conversion is quite sharp at the electrical biasing case (Fig. 4c, d and Suppl. Fig. 12). The difference in the interface structure in these two experiments may be caused by different heating treatments. As current flows through the converted NWs region and induces local joule heating, the electrical biasing may engender more localized defect generation at the interface, while direct heating can be considered as more global. We also find that changing the bias voltage and heating temperature (above threshold value) will only result in altering the growth speed of NWs, as illustrated in Suppl. Fig. 16, higher temperature will have faster growth rate than the electrical bias experiment (detailed growth rate shown in Suppl. Figs. 15e, f). Without graphite confinement, the $MoTe_2$ film turns into a layer of mixed disordered NWs and Mo clusters upon electrical biasing (Fig. 3b) or heating (Suppl. Fig. 14), consistent with ref.[21].

The formation of the $MoTe_2$-graphite interface in the heterostructure is also crucial in this transition. Firstly, after fabricating the graphite-encapsulated $MoTe_2$ heterostructure, two metal-semiconductor Schottky barriers form at the contact interfaces due to the different work function of graphite and $MoTe_2$[27–31]. The applied electrical bias will therefore, induce a forward and reverse bias on the two Schottky barriers and generate Joule heating at the reverse-biased interface. In this case, the Te vacancies can then be generated in the

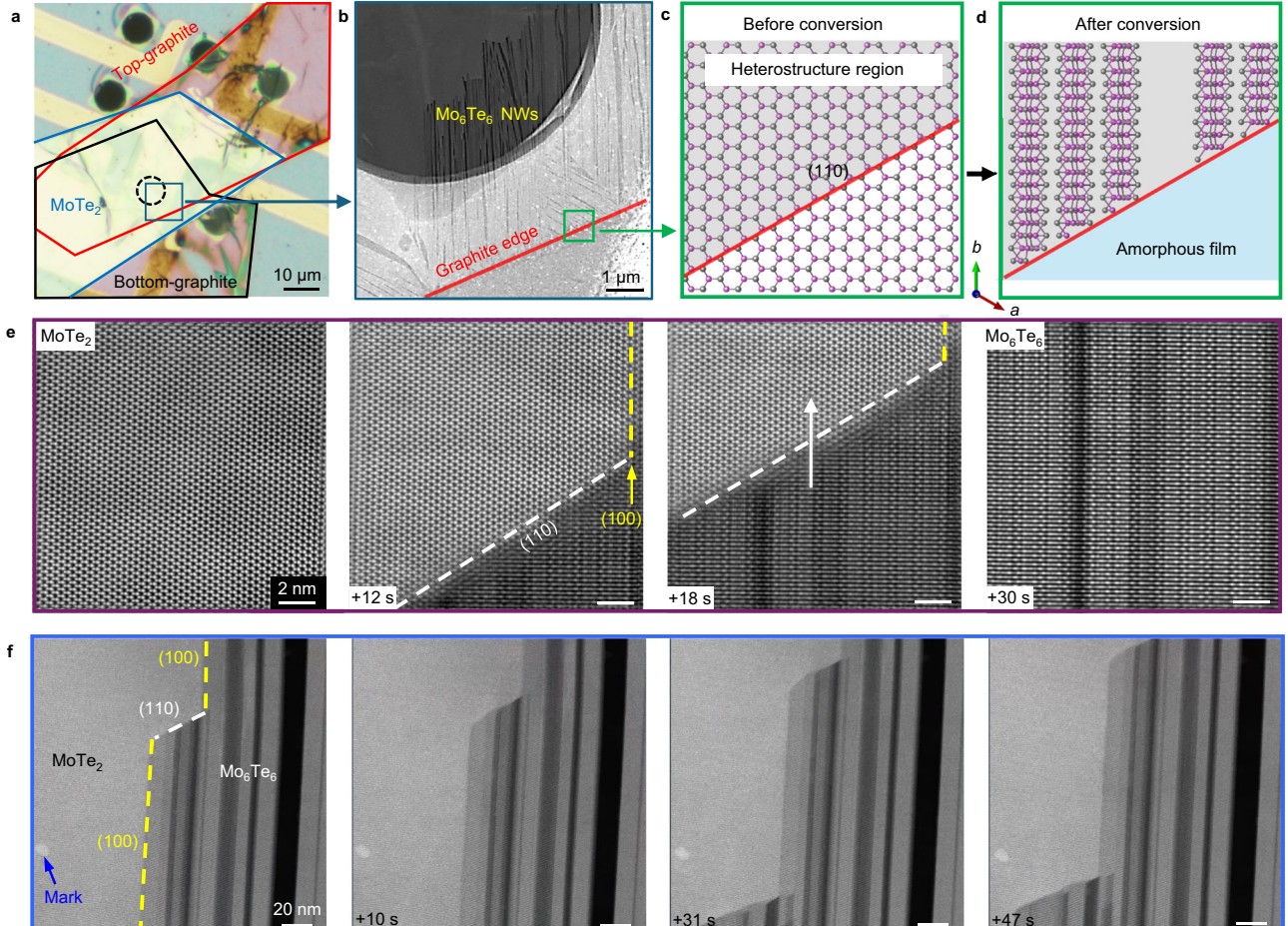

**Fig. 3 | In-situ dynamical process of the highly oriented $Mo_6Te_6$ NWs formation under electrical bias. a** Optical microscope image shows the geometry of the graphite sandwiched $MoTe_2$ heterostructure. The top and bottom graphite are marked by red and black outlines, respectively, while the $MoTe_2$ film in between is marked by a blue outline. Highly oriented growth of NWs was observed in the freestanding hole on the chip indicated by a dashed black circle. **b** Zoom-in low-magnification HAADF-STEM image taken at the heterostructure region where $MoTe_2$ is covered by graphite on both sides, within the area indicated by the blue box in (**a**). Bundle-like structure, namely $Mo_6Te_6$ NWs, is seen in the region that is fully sandwiched between the graphite sheets (above the red line). Below the red line, absent the graphite sheets, $MoTe_2$ is converted to amorphous film composed

of random size clusters. **c, d** Schematics of the structural transition from the graphite confined $2H\text{-}MoTe_2$ to $Mo_6Te_6$ NWs before and after applying electrical biasing within the area indicated by the green box in (**b**). The area shaded in grey is the heterojunction region with the graphite edge marked by a red line. The $MoTe_2$ film that is not sandwiched in graphite sheets in the blue area of (**d**) is converted into amorphous film composed of clusters. Series of (**e**) high- and (**f**) low-magnification HAADF-STEM images that taken during the conversion of $2H\text{-}MoTe_2$ into $Mo_6Te_6$ NWs. The $MoTe_2$ (110) and (100) facets are highlighted by white and yellow dashed line in (**e**) and (**f**), and the growth direction of NWs is highlighted by white arrow in (**e**).

part of the $MoTe_2$ layer that is close to the lower graphite sheet when a critical bias is reached (sufficiently high temperature by Joule heating). The formation of vacancies is confirmed by the presence of complex defect patterns at the growth frontier at the $MoTe_2$ side near the $MoTe_2$/NWs interface, as shown in Fig. 4e. Such complex defect patterns have been shown to be generated when Te vacancies are created by vacuum annealing[32]. The corresponding NWs growth schematic induced by Joule heating is shown in Suppl. Fig. 16. It is also notable that unlike prior work that obtained transformations of $MoTe_2$ to other phases by vacancy generation[8,18,21,26,33–37], we only observe the direct transformation to oriented NWs, most likely because of the unique heterostructure we use to confine the material.

It seems that vertical conversion of NWs is more thorough in the electrical biasing case (detailed discussion in Suppl. Fig. 16), as a sharp $MoTe_2$/NWs connected interface is always present (Figs. 3e and 4c, d), while an overlapping interface is persistently seen in the case of global heating (Figs. 4a, b), presumably due to localized vs. global heating conditions. We also notice that in either electrical biasing or confined heating experiments, no particles or clusters of the escaped Te atoms are observed. We infer that, when the escaped Te atoms enter the gaps

(NWs region), they are more likely to migrate along the gap and rapidly vaporize from the heterostructure inside the ultra-high vacuum of the microscope.

To further elucidate the defect-induced formation of NWs, molecular dynamics (MD) simulations based on density functional theory (DFT) are performed. Figure 4f, g shows the process of simulating the $Mo_6Te_6$ NW formation in which a single NW is placed against a $2H\text{-}MoTe_2$ monolayer (see Suppl. Fig. 17 for a single NW facing a slanted zigzag edge). To simulate the Te defects generated by the bias voltage during experiments, we artificially removed 50% of the Te atoms in $MoTe_2$. After that, by placing the structure at 1000 K, after 10 ps, a $Mo_3Te_3$ ring is found at the head of the pre-placed NW, as shown in Fig. 4g. This MD result indicates that under the conditions of heating and high concentrations of missing Te atoms, NWs can grow along the $MoTe_2$ vertical zigzag direction (red lines in Fig. 4f). The slanted zigzag edge (blue lines in Fig. 4f) is part of the growth process because it is the energetically favoured edge, as revealed by DFT calculations (the energy density is 0.26 $eVA^{-1}$ for zigzag edges but 0.40 $eVA^{-1}$ for armchair edges).

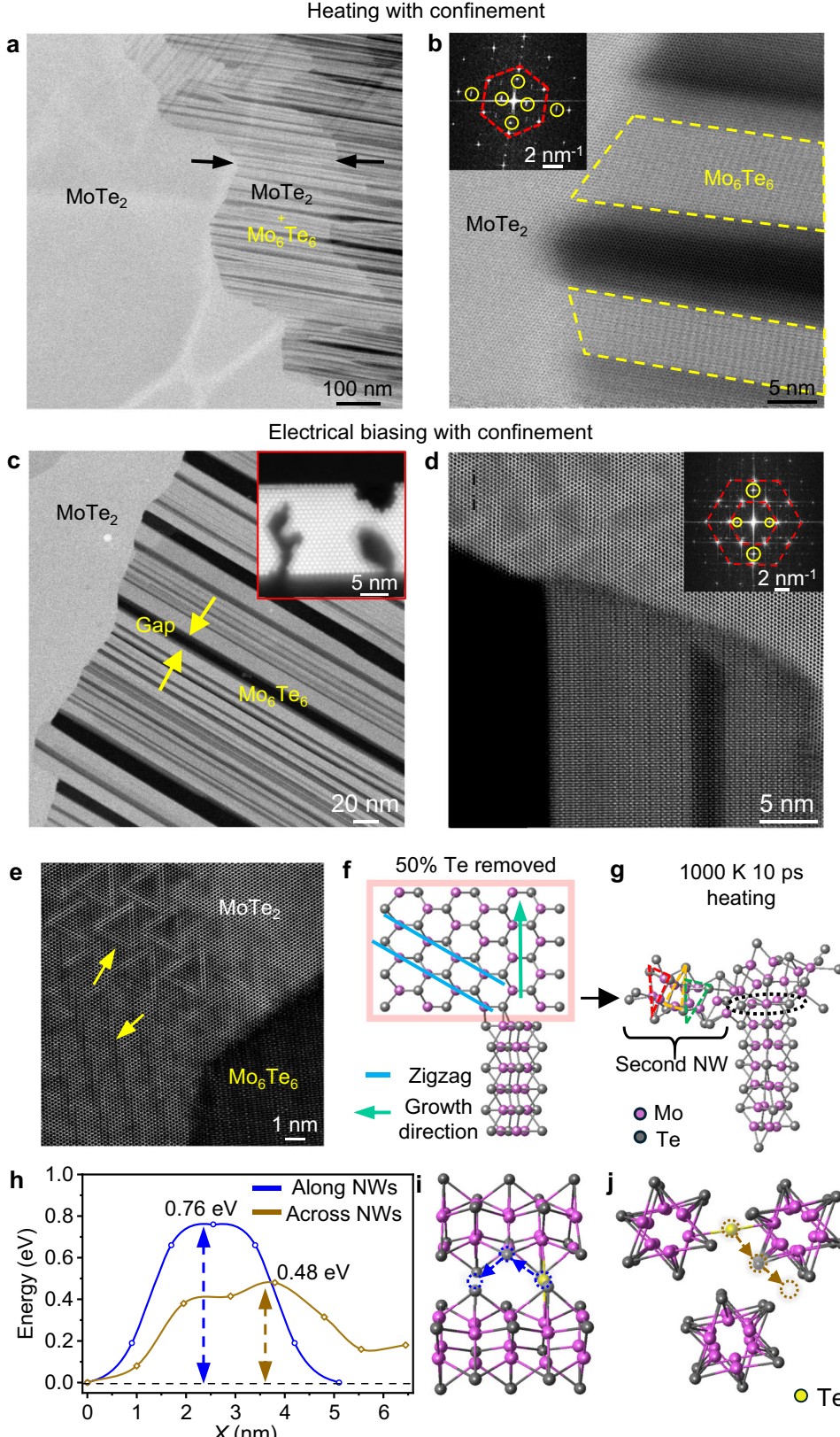

Figure 4h present potential energy barriers for the migration pathways shown in Fig. 1c. The detailed schematic regarding these two migration paths is shown in Fig. 4i, j. The diffusion energy barrier for Te atoms migrating along NWs (Fig. 4i) is 0.76 eV (0.69 eV on a single NW, Suppl. Fig. 18) and 0.48 eV (Fig. 4j) for migrating laterally to the gaps, which is even lower than along the NWs. These relatively small energy barriers (<1 eV) enable migration at room temperature and suggest that, even more so in the heated environment of the experiment, the ejected Te atoms tend to migrate laterally and finally accumulate within the void spaces among NW bundles. When Te atoms enter the gaps, they may either condensate to form droplets or wet the void spaces. DFT calculations find that the total energy of Te adhere to a

**Fig. 4 | Growth mechanism of oriented NWs by controlled in-situ experiments.** Low (**a** and **c**) and high-magnification (**b** and **d**) HAADF-STEM images showcasing the conversion of $Mo_6Te_6$ NWs from 2H $MoTe_2$. **a,b** 2H phase region overlap can be observed in heating with confinement, with the FFT pattern inserted in (**b**) (the FFT patterns highlighted by red dash line and yellow circle representing the 2H $MoTe_2$ and $Mo_6Te_6$ NWs). **c,d**. The interface structure in electrical biasing is clean and sharp. The inserted images in (**c**) showcase the cross-sectional HAADF-STEM images of the gap structure in the NW region. The inserted image in (**d**) is the FFT pattern corresponding to the STEM image in (**d**). The FFT pattern regarding to 2H $MoTe_2$ and $Mo_6Te_6$ NWs are highlighted by red dash lines and yellow circles. **e** High-resolution STEM image showing triangular- and line-shaped defects formed in the $MoTe_2$ region (yellow arrows) near the growth frontier during electrical biasing. **f–j** MD simulation of the $Mo_6Te_6$ NW formation in a $MoTe_2$ lattice with Te deficiency. **h** DFT-calculated energy barrier for a Te atom migrating along and across three close-packed $Mo_6Te_6$ NWs (blue and brown line highlighted). The migration energy barriers are calculated by changing the x-axis coordinate of the Te atoms. (x= The horizontal coordinate position of the Te atom relative to its starting point along the migration path, with the origin set at the initial position of Te in the supercell). The corresponding schematics are shown in **i,j**. Source data are provided as a Source Data file.

---

single NW is 0.5 eV higher than that of Te in the bulk, signifying a repulsive interaction with the NWs. Considering that the growth of NWs originates from the edge of the heterostructure, these gaps can connect the newly exposed $MoTe_2$ edge to the vacuum chamber in the STEM, enabling Te atoms to migrate freely along the gaps and away from the heterostructure.

### Contact properties of $Mo_6Te_6$ NW based devices

An isolated $Mo_6Te_6$ NW is an indirect bandgap semiconductor[13], while bulk $Mo_6Te_6$ NWs exhibit a partially occupied conduction band, indicating their conductive nature as proper contact candidates to TMDs devices[21]. Using the same confined strategy revealed by the atomic STEM results, we directly pattern NWs on devices and measure their contact performance. Figure 5a shows the linear $I–V$ curve of the patterned NW bundles, and the inset of $R–T$ measurement confirms its metallic characteristic. We deposit metal electrode arrays on the patterned $Mo_6Te_6$ NWs to measure the contact resistance at the metal-NW interface. We choose four metal deposition recipes to estimate the conducting nature of the NWs as contact. Figure 5b illustrates the linear curves of resistance vs. channel lengths at interfaces between NWs and Au + Ti, Au + Cr, Cr + Ti, and Pd + Ti electrodes, where Ti or Cr often serve as adhesion layers with a thickness of approximately 6 nm[38,39]. Among these four combinations, gold electrodes exhibit minimal contact resistance ($\approx 43.7 \,\Omega\,\mu m$ for Au + Cr and 88.4 $\Omega\,\mu m$ for Au + Ti). The low contact resistance is possibly attributed to the conductive nature of the NWs and the close contact formed between the deposited metal electrodes and the NWs (bundle structure with random gaps) compared to traditional metal-2H or metal-1T' contacts.

Next, we have fabricated NWs-$MoTe_2$-NWs FETs and measured its transport properties based on the schematic shown in Fig. 5c (the detailed device preparation is shown in "Method" section). The corresponding OM images of the heterostructure and the fabricated devices are presented in Fig. 5d, e. In Fig. 5f, the corresponding gate-voltage-dependent $I_{ds}$-$V_{gs}$ curve [i.e., drain–source current ($I_{ds}$) versus gate voltage ($V_{gs}$)] exhibits intriguing ambipolar characteristics under gate voltage modulation. In addition, $I_{ds}$-$V_{ds}$ curve measurements (shown in the insert in Fig. 5f) reveals a linear behavior with a resistance of approximately 0.9 MΩ [i.e., drain–source current ($I_{ds}$) versus drain–source voltage ($V_{ds}$) at $V_{gs} = 0$]. In contrast, $MoTe_2$ devices contacted directly with graphite electrodes (shown in Suppl. Fig. 19) or vertical stacked NWs[16] show non-linear behavior. The carrier mobility is calculated to be 5.99 $cm^2V^{-1}s^{-1}$. To determine the Schottky barrier height $q\Phi_B$, it is common to use an Arrhenius plot, i.e., ln ($I_{ds}/T^{3/2}$) against $1000/T$ for various $V_{ds}$. The Arrhenius graphs for the NWs-$MoTe_2$-NWs device at $V_{gs} = 0$ V with temperature from 200 to 300 K are shown in Fig. 5g. The slope S was extracted as a function of $V_{ds}$, where the intercept $S_0 = -\frac{q\phi_B}{1000 \, k_B}$ can be used to further determine the Schottky barrier height (Fig. 5h). The extracted Schottky barrier height in our case is 11.5 meV.

We have compared our transport results with previous reported works using the 1T' phase as contact, as shown in Fig. 5i and Suppl. Table 1. In our case, comparing with the contact resistance between 1T'/NWs and metal electrodes, the contact resistance of our results is the lowest among previous findings (43.7 $\Omega\,\mu m$)[8,9,16,18,21,36,40–44]. The Schottky barrier is nearly halved compared with 1T' contact, and

compatible to the previously reported vertically stacked Pd/$Mo_6Te_6$ NWs/2H-$MoTe_2$ back-gated FET where $q \,\Phi_B = 8.7$ meV and $R_c = 28.7$ MΩ[16]. The small contact resistance at the metal-NW interface and relatively low Schottky barrier in the NW-2H contact FET device demonstrates the potential of NWs that can serve as promising contacts in nanodevices.

## Discussion

In summary, we proposed a strategy for patterning highly oriented $Mo_6Te_6$ NWs in 2D $MoTe_2$, which was demonstrated by applying in-situ electrical bias to a graphite-encapsulated 2D $MoTe_2$ heterostructure in a STEM. Through dynamical in-situ studies and DFT calculations, we find that the conversion of directional NWs growth originated from both the confinement effect of the graphite electrodes and the local Joule-heating generated by the applied bias, resulting in a sharp metal-semiconductor interface. Moreover, the low zigzag-edge energy density of 2H $MoTe_2$ and the anisotropic Te atom migration paths together contribute to the emergence of highly oriented growth of NWs. Furthermore, using the same strategy to directly pattern oriented NWs on 2D $MoTe_2$ devices, we demonstrated that the introduction of NWs can significantly enhance the performance of device contact. Our work introduces the potential that NWs can serve as interconnections in flexible 2D nanoelectronics.

## Methods

### Sample preparation

Few layers 2H phase $MoTe_2$ and graphite layers (HQ Graphene), were exfoliated onto the silicon wafer (300 nm thick $SiO_2$) from bulk crystals using scotch tape cleavage method. Next by using the 2D-transfer platform and self-developed Polycarbonate (PC) micro-dome dry transfer method. PC film was made by dissolving PC particles (Alfa Aesar) into chloroform (Alfa Aesar) at 6% mass fraction. Then the solution was dropped onto a glass slide and covered with another glass slide. After the chloroform evaporated, PC film was obtained. The detailed schematic of the PC-dome can be seen at Suppl. Figure 2. We stacked graphite and $MoTe_2$ in sequence to form a sandwiched heterostructure. Then we dropped down the heterostructure onto a 4-pin MEMS chip made by Protochips through heating the PC film at 180 °C. To avoid contamination and sample oxidation, we performed all the transfer processes in nitrogen-filled glove boxes. Before each in-situ experiments, we performed plasma cleaning to reduce the carbon deposition during imaging.

### In-situ STEM measurements

The in-situ microscopy characterization experiment was conducted in a commercialized STEM (FEI Titan Themis G2) at 300 kV accelerating voltage. A double spherical-aberration corrector (DCOR) and a high-brightness field-emission gun (X-FEG) with a monochromator is installed onto this microscope, so that the obtained images have a relatively high resolution. The inner and outer collection angles for the STEM images ($\beta$1 and $\beta$2) were 48 and 200 mrad, respectively, with a semi-convergence angle of 25 mrad. To introduce electric bias during electron microscope characterization, we select a special TEM holder (Protochips Aduro 300) equipped with special MEMS chips. This

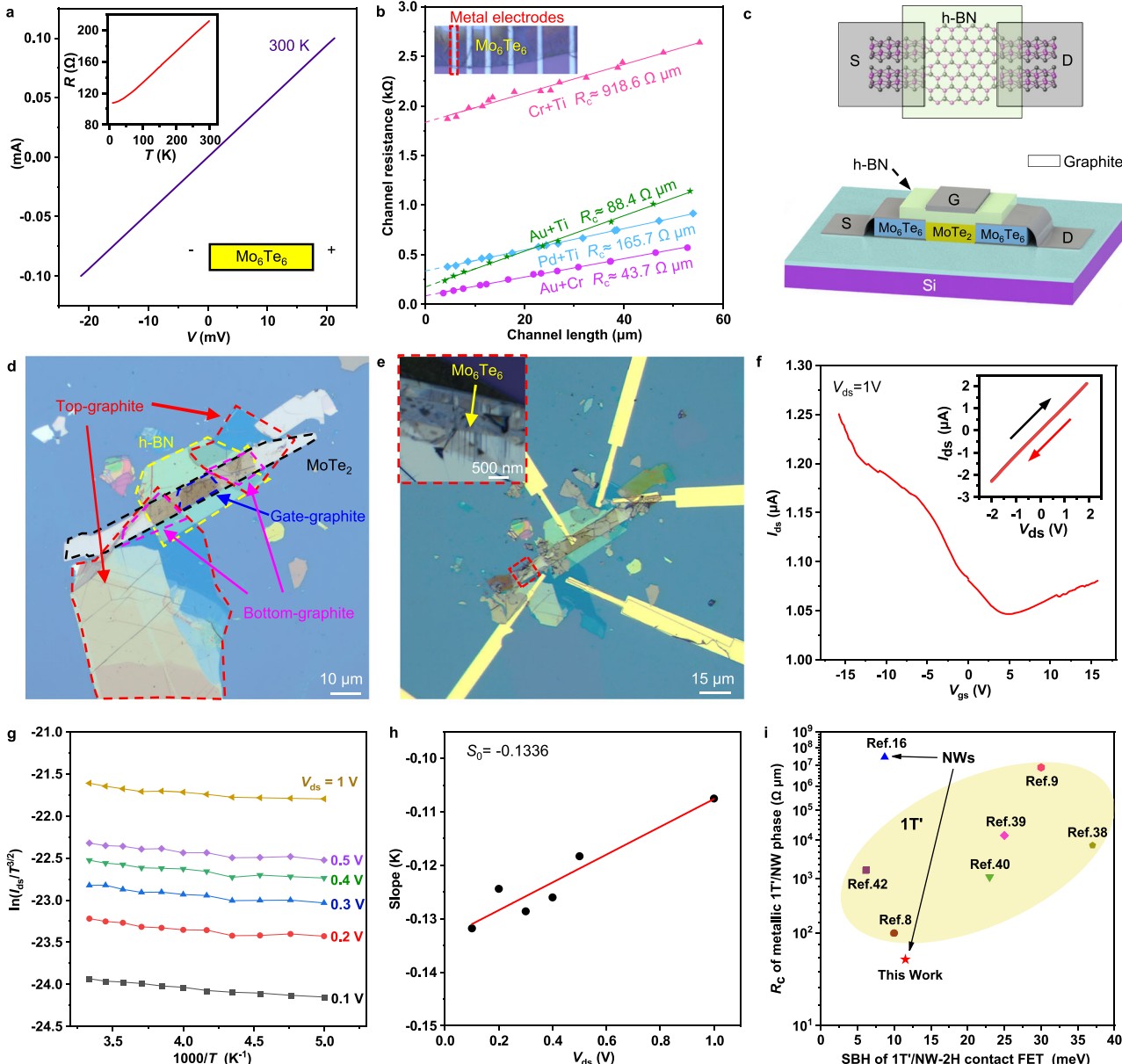

**Fig. 5 | Transport results of Mo₆Te₆ NWs and fabricated (NW-MoTe₂-NW) FET device. a** Transfer curve ($I_{ds}$−$V_{ds}$) of Mo₆Te₆ NWs device, showing linear behaviour. The inserted $R$-$T$ curve demonstrates the metallic nature of bulk Mo₆Te₆ NWs. **b** Contact resistance between Mo₆Te₆ NWs and Au+Ti, Au+Cr, Cr+Ti, and Pd+Ti metal electrodes. The Ti and Cr metal are deposited as adhesion layer with thickness around 6 nm. **c** Schematic of our NWs-MoTe₂-NWs FET device. **d** Optical-microscope images of the fabricated graphite confined MoTe₂ heterostructure, and (**e**) the fabricated FET device. We introduce graphite confinement and fabricated oriented NWs at both ends of the MoTe₂. An optical-microscope image confirming the conversion of oriented NWs at the graphite edge is shown in the inserted image in (**e**), (**f**) Transfer curve ($I_{ds}$−$V_{gs}$) at $V_{ds}$ = 1 V, with $I_{ds}$−$V_{ds}$ characteristics of the

device at room temperature with gate voltage ($V_{gs}$) = 0 inserted, showing linear behaviour. **g** Arrhenius plot ln($I_{ds}/T^{3/2}$) versus 1000/$T$ at different values of $V_{ds}$ ($T$ vary from 300−200 K). **h** Extraction of $q\Phi_B$ via the intercept value, where each data point represents the slope obtained from the Arrhenius plot in (**g**) using a specific value of $V_{ds}$. **i** Comparison of contact resistance between 1 T'/NWs-metal electrodes and Schottky barrier height of the 1 T'/NWs-2H contact FET of this work with the previous reported results[8,9,16,18,21,36,40–44]. The devices with 1 T'−2H MoTe₂ contact are highlighted in the yellow ellipse, while NW-2H contact devices are highlighted by black arrows. See Suppl. Table 1 for a detailed comparison. Source data are provided as a Source Data file.

MEMS chip can apply bias and heating to the sample during the imaging process. A power supply (2616 A System Sourcemeter) and software controller (Fusion 350) were linked to the holder for controlling the sample temperature.

**DFT calculations and MD simulations**
DFT calculations were done using the projector augmented-wave method as implemented in the VASP (Vienna Ab initio Simulation Package) code. We employed the Perdew-Burke-Ernzerhof version of

the generalized gradient approximation to the exchange-correlation functional. The Grimme-D 2 correction was included to take the van der Waals interactions into account. Total energy calculations were done with an energy cut-off of 450 eV, while the energy cut-off is set to 350 eV in molecular dynamics simulations. The MD simulations were carried out with the temperature set to 1000 K. The minimum path of potential energy surface was calculated using the nudged-elastic band method as implemented in VTST (the transition state tools for VASP) code.

## NWs-metal electrodes contact device fabrication

First few layers of $MoTe_2$ were exfoliated onto the silicon wafer from bulk crystals using scotch tape cleavage method. The thermal annealing was applied (730 °C for 5 min) to introduce $Mo_6Te_6$ NWs from the $MoTe_2$ flake. Subsequently, we used standard UV lithography and electron beam evaporation techniques to deposit metal electrodes with a width of 3 μm on the converted $Mo_6Te_6$ regions. For the deposition sequence and relative thickness of the metal electrodes, please refer to the main text. We used Keithley 2400 SMU to measure the device current by changing voltage.

## NWs-MoTe₂-NWs FET device fabrication

Few layers of graphite, $MoTe_2$, and h-BN were exfoliated onto the silicon wafer from bulk crystals using scotch tape cleavage method. Next, by using the 2D-transfer platform and self-developed PC micro-dome dry transfer method, graphite layers were stacked at both ends of the $MoTe_2$ flake, forming two heterostructure. Then, top-gate graphite and h-BN dielectric layers were picked up in sequence and dropped down on top of the $MoTe_2$ (sealed the exposed $MoTe_2$ in the middle, leaving $MoTe_2$ at both ends exposed). Then, we used thermal annealing at 730 °C for 5 min to convert oriented NWs at both ends of the $MoTe_2$. The Au-Ti electrodes are deposited on graphite layers through standard UV lithography and electron beam evaporation, forming FET devices.

## Carrier motilities and Schottky barrier height measurements

We used Keithley 2400 SMU to measure the drain-to-source current ($I_{ds}$) by biasing the drain-to-source voltage ($V_{ds}$), while gate voltages were applied using another Keithley 2410 SMU. The carrier motilities of the NW-MoTe₂-NW FETs can be extracted from the from Eq. (1)[16]:

$$\mu = \left(\frac{L}{W}\right)\left(\frac{d}{\varepsilon_0 \varepsilon_r V_{ds}}\right)\left(\frac{dI_{ds}}{dV_{gs}}\right) \tag{1}$$

where a channel length ($L$) of 140.72 μm, width ($W$) of 19.63 μm, dielectric thickness ($d$) of 200 nm, permittivity ($\varepsilon_r$) of 3.4, and ($dI_{ds}/dV_{gs}$) of $-1.26 \times 10^{-8}$ A V$^{-1}$ and $3.36344 \times 10^{-9}$ A V$^{-1}$ extracted from a linear fit of the transfer curve ($I_{ds}$–$V_{gs}$ at a $V_{ds}$ value of 1 V), are used in the calculation. The carrier mobility is calculated to be 5.99 cm2 V$^{-1}$ s$^{-1}$ for holes, and 1.56 cm2 V$^{-1}$ s$^{-1}$ for electrons. The drain-source current $I_{ds}$ can be defined by the 2D thermionic emission Eq. (2)[45]:

$$I_{ds} = A^*_{2D} S T^{3/2} \exp\left[-\frac{q}{k_B T}\left(\phi_B - \frac{V_{ds}}{n}\right)\right] \tag{2}$$

where $A^*_{2D}$ is the 2D equivalent Richardson constant, $n$ is the ideality factor, and $V_{ds}$ is the drain-source bias voltage. To determine the Schottky barrier height $\Phi_B$, it is common to use an Arrhenius plot, i.e., ln ($I_{ds}/T^{3/2}$) against 1000/$T$ for various $V_{ds}$. The slope $S$ in the Arrhenius graphs for the NWs-MoTe₂-NWs device at $V_{gs} = 0$ V from 200 to 300 K can be extracted as a function of $V_{ds}$, where $S = -(\frac{q}{1000 k_B})(\phi_B - \frac{V_{ds}}{n})$. The Slope $S$ can be plotted against $V_{ds}$, then after acquired the intercept $S_0 = -\frac{q\phi_B}{1000 k_B}$, the Schottky barrier height can be determined.

## Reporting summary

Further information on research design is available in the Nature Portfolio Reporting Summary linked to this article.

## Data availability

The data that support the findings of this study are available from the corresponding authors upon request. Source data are provided in this paper.

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

## Acknowledgements

This work was supported by the National Natural Science Foundation (12104206, 11974156); Guangdong Innovative and Entrepreneurial Research Team Program (Grant No. 2019ZT08C044), Guangdong Basic Science Foundation (2023B1515120039), Shenzhen Science and Technology Program (No. 20200925161102001), the Science, Technology, and Innovation Commission of Shenzhen Municipality (No. ZDSYS20190902092905285), and Quantum Science Strategic Special Project from the Quantum Science Center of Guangdong-Hong Kong-Macao Greater Bay Area (GDZX2301006). STEM characterization was performed at the Pico Center from SUSTech Core Research Facilities that receives support from the Presidential Fund and Development and Reform Commission of Shenzhen Municipality. Theoretical work and data analysis at Vanderbilt University was supported in part by the U.S. Department of Energy, Office of Science, Basic Energy Sciences, Materials Science and Engineering Division Grant No. DE-FG02-09ER46554 and by the McMinn Endowment.

## Author contributions

J.L. conceived the idea and supervised the project. Q.Y., L.Z., X.L., designed and performed the in-situ STEM experiments. S.T.P., Q.Y., and L.Z. analyzed the data. S.T.P. and Y.W. participated in the theoretical research. Q.Y., K.L, and Y.C. designed and conducted the transport device experiment. Y.W. performed MD simulation and DFT calculation. E.Z. provided the schematic image for the experiments. Q.Y., X.S., Z.C., J.Z., S.T.P., J.L. drafted the manuscript. All authors contributed to data analysis, result interpretation, and writing of the manuscript.

## Competing interests

The authors declare no competing interests.
