## [Peer Review File · Nature Communications]

Constrained patterning of orientated metal chalcogenide nanowires and their growth mechanismREVIEWER COMMENTS

Reviewer #1 (Remarks to the Author):

In this manuscript, the authors report that 1D Mo₆Te₆ nanowires with specific directions can be formed by applying electrical bias to graphite-sandwiched 2D MoTe₂. They used in-situ STEM to support these results and performed DFT calculations to propose the growth mechanism of the 1D Mo₆Te₆ nanowires. Although the formation of 1D Mo₆Te₆ from 2D MoTe₂ by the generation of Te vacancies in 2D MoTe₂ was already reported by several different research groups, the authors discovered that the 1D Mo₆Te₆ nanowires can grow directionally through the confinement effect induced by graphite layers. Although the results in the manuscript are interesting, there are some issues that have to be considered. I believe that the manuscript can be reconsidered if the following issues are resolved.

1. The authors claimed that their results can be utilized for nanoelectronics. To support this, it would be great if they could fabricate the devices based on oriented Mo₆Te₆ nanowires or analyze the characteristics of the junction between oriented Mo₆Te₆ nanowires and 2D MoTe₂.
2. The characterization of the 1D Mo₆Te₆ nanowires and 2D MoTe₂ is incomplete. In order to confirm the formation of materials, optical and crystallographic evidences have to be provided. Although the authors provided STEM and EDS data to characterize the 1D Mo₆Te₆ nanowires and 2D MoTe₂, I do not see the optical and crystallographic data such as Raman spectra, SAED patterns, and XRD patterns.
3. They claimed that the origin of the phase transition is the formation of Te vacancies induced by Joule-heating. If the Te atoms were released from MoTe₂, there should be particles or clusters of Te atoms. Were you able to observe any of these?
4. In this paper, the authors claimed that 2D 2H MoTe₂ is directly converted to 1D Mo₆Te₆. However, there is a paper (Nano Lett. 2018, 18, 2, 675–681) that reports that 1T' MoTe₂ could be generated in addition to 1D Mo₆Te₆ during the transition from 2H MoTe₂ to 1D Mo₆Te₆ induced by the formation of Te vacancies. In addition, both experiments and

calculations have reported that the phase transitions from 2H MoTe₂ to 1T' MoTe₂ occur as the concentration of Te vacancies increases, (Nat. Commun., 2014, 5, 4214; Science, 2015, 349, 625-628; Adv. Mater. 2017, 29, 1605461). Is there a possibility that 2H MoTe₂ can be converted to 1T' MoTe₂ in the authors' work?

5. How thick is the graphite layers? Are there any possibility that the results will vary depending on the thickness of graphite layers?

6. In Figure 3, low-magnification STEM images and some low-quality high-magnification STEM images are not enough to confirm the formation of 1D Mo₆Te₆ nanowires from 2D MoTe₂. The authors should provide additional data.

Reviewer #2 (Remarks to the Author):

Report for "Direct observation of controlled growth of oriented metal-chalcogenide nanowires " by Q. Yang et al.

The authors report on growth of highly oriented Mo₆Te₆ nanowires. The conversion of Mo₆Te₆ NWs from MoTe₂ was thoroughly studied by STEM and first-principles calculations, which reveals that Te vacancy formation is the key. The characterization and explanation of the fabrication and properties of Mo₆Te₆ NWs were made in a convincing way. However, I do not recommend publication of this manuscript by its novelty and impact issues:

1. The impact and motivation of this work are vague. The authors claimed that this kind of TMC-NWs would serve as conductive interconnections and decrease the Schottky barrier height in FETs. However, interconnect works in semiconductor industry and nanodevice engineering communities are not actively using these materials.

As for the Schottky barrier height study, there have been lots of studies using phase engineering (based on 1T-2H transition in MoTe₂), which is more versatile than this one; laser irradiation, strain, intercalation, and other methods enable this. To overcome this issue, the authors should provide more solid and quantitative comparison data with their

Mo₆Te₆ and other MoTe₂ studies.

2. Many groups have reported the synthesis of similar TMDC nanowires (some of them are cited in this manuscript). And it is hard to see the advantages or novelty of Mo₆Te₆ NWs over them. Below are some details:

(1) One paper (NanoLett.2020, 20, 8866) suggests W₆Te₆ nanowires show unique features such as semiconductor to metal transition, Luttinger liquid behaviors, which are based on its 1D confined geometry. This would be a good reason to use the nanowire. Are there similar benefits (e.g., fabricating 1D-2D-1D heterostructures) with Mo₆Te₆?

(2) The disordered Mo₆Te₆ NWs (Fig. 3c) show different transport? Some unique features like transport should be shown in this work for high impact journals.

Reviewer #3 (Remarks to the Author):

The manuscript describes the oriented growth of Mo₆Te₆ nanowires (NWs) from a graphite-encapsulated MoTe₂ heterostructure driven by local Joule-heating induced by electrical bias, using in-situ Scanning Transmission Electron Microscopy (STEM), Density Functional Theory (DFT) calculations, and Molecular Dynamics (MD) simulations. It shows a detailed analysis of nanowire growth. However, the submitted manuscript does not provide any new significant research findings compared to what's been published by other researchers. In particular, the exhibited results are very similar to the one by Ref. 14 (Zhu et al., Adv. Mater. 2017, 29, 1606264). The only difference is whether the main activation energy for nanowire phase transformation is provided by direct heating or joule heating induced by electrical biasing. And yet, no estimate is given for the joule heating temperature induced by electrical bias. Other findings are almost identical, such as the metallic nature of bundled nanowires vs. the small gap semiconductor nature of an isolated nanowire; a sharp interface between Mo₆Te₆ nanowire and 2D MoTe₂; intercalated chalcogen atoms along the gap between nanowires in transition to a stable energy minimized configuration. The highly oriented growth of nanowires presented in the submitted manuscript is noteworthy, though.

The following couple of general comments are provided for the authors to consider:

1. The electrical in-situ STEM experiment is based on the heterostructure attached to MEMS chips. The vertical field is achieved by the top and bottom graphite layers in the heterostructure are connected to the electrodes. In Fig.2 (a), and Fig. S1, the bottom layer graphite (red marked) seems to connect with negative electrode as expected, and the metal wire linked to the positive electrode as well. However, the resistance vs time (Fig.1 c) does not show the short feature. Any efforts help with insulation and prevent short? Could the author provide more details on the MEMS circuit and the current path?
2. The HAADF image exhibits fringe contrast (Fig.1 (b.i)), how to interpret the contrast? Is there any ripple feature on the heterostructure that may affect the strain & stress of materials? A micrometer scale cross-section image or AFM data on the heterostructure would help to better understand the graphite-sandwiched MoTe₂ structure.
3. The authors attribute the formation of the overlapping interface is caused by the temperature gradient. What is the thickness of the MoTe₂ layer? Could the author give an estimation of the temperature difference between top and bottom? The temperature difference might not be significant at the nanoscale dimension.

The author compared the result between in-site anneal and in-site electric STEM experiment. And conduct the electrical biasing case may engender more defect that result into an intense transformation and form a sharp interface. However, the in-situ experiment on heterostructure is complex that the authors need to provide more evidence to convince reader.

1. What the is thickness of the Graphite layer and MoTe₂ layer used in both experiments?
2. Is the interface structure between MoTe₂ and NWs bias or temperature related? Or ramping related?
3. Any direct comparison with anneal experiment, that exhibit electrical experiment is “more intense?” Like serial images showing phase transformation area vs time?
4. The author shall clearly indicate the advance or the uniqueness of the proposed electrical induced phase transformation strategies.

The concept of electrical induced phase transformation and confinement effect on MoTe₂

material has been widely discussed in published works (Nature, 2017, 550(7677): 487-491, Ref 18 in this manuscript) (Nature materials, 2019, 18(1): 55-61) (Advanced Functional Materials, 2021, 31(51): 2107376) (Nano letters, 2023, 23(2): 677-684). I would propose to discuss these studies, and highlight the innovation and significance of the present work.

Reviewer #4 (Remarks to the Author):

Response to reviewers' comments (NCOMMS-23-41661A)

Title: Direct observation of controlled growth of oriented metal-chalcogenide nanowires

Authors: Qishuo Yang, Yun-Peng Wang, Xiao-Lei Shi, XingXing Li, Erding Zhao, Zhi-Gang Chen, Jin Zou, Kai Leng, Yongqing Cai, Liang Zhu,* Sokrates T. Pantelides, Junhao Lin*

We sincerely thank the reviewers for the constructive comments on our manuscript. We have addressed all the comments point-by-point and revised the manuscript accordingly. In this Response Letter, comments from the referees are in **regular black** typeface, our responses are in regular **blue** typeface. All major changes have been highlighted in **red** in the main text and supplementary information (SI). A list of the main changes in the manuscript and SI is provided below:

1. New series of atomic-resolved HAADF-STEM images are provided to confirm the conversion of NWs in the electrical biasing with confinement, heating with confinement, and heating without confinement experiments. We also provide additional cross-sectional STEM data regarding to the thickness analysis of the confined graphite layers and the encapsulated MoTe₂ layers.
2. We also fabricated NWs-MoTe₂-NWs FET devices through direct NWs patterning on MoTe₂, demonstrating that the introduction of NWs can enhance the device contacts.
3. We have added new references and rephrased the dissuasions to provide a more convincing motivation regarding the NW applications as suggested by the reviewers.
4. We provide more solid and quantitative comparison data with the Mo₆Te₆ NWs and other MoTe₂ studies to show the scientific advancement of our work (see Table in the response).
5. We rephased the corresponding description regarding the comparison between local joule heating generated by electrical biasing and global heating.
6. We added Prof. Kai Leng and Prof. Yongqing Cai as new co-authors for their constructive ideas in designing the prototype devices and fruitful discussions of the manuscript.

We sincerely express our gratitude once again to the reviewers for your valuable time, and we believe that all your concerns have been addressed in this response.

Reviewer #1 (Remarks to the Author):

In this manuscript, the authors report that 1D Mo₆Te₆ nanowires with specific directions can be formed by applying electrical bias to graphite-sandwiched 2D MoTe₂. They used in-situ STEM to support these results and performed DFT calculations to propose the growth mechanism of the 1D Mo₆Te₆ nanowires. Although the formation of 1D Mo₆Te₆ from 2D MoTe₂ by the generation of Te vacancies in 2D MoTe₂ was already reported by several different research groups, the authors discovered that the 1D Mo₆Te₆ nanowires can grow directionally through the confinement effect induced by graphite layers. Although the results in the manuscript are interesting, there are some issues that have to be considered. I believe that the manuscript can be reconsidered if the following issues are resolved.

Reply 1-0: We thank the reviewer for the concise summary of our work and appreciate the comments that further improve the manuscript. We have prepared a point-to-point response to address all the reviewer's comments.

Comment 1-1. *The authors claimed that their results can be utilized for nanoelectronics. To support this, it would be great if they could fabricate the devices based on oriented Mo₆Te₆ nanowires or analyze the characteristics of the junction between oriented Mo₆Te₆ nanowires and 2D MoTe₂.*

Reply 1-1: Thank you for this valuable suggestion. We have developed prototype transport devices to demonstrate the potential applications of the oriented Mo₆Te₆ nanowires (NWs) in nanoelectronics. Using similar fabrication strategy of *in-situ* devices for TEM experiments, we first fabricated oriented NWs at both ends of the MoTe₂ flake, which is realized by the oriented growth of NWs from the edges of the double-sided graphite confinement, with subsequent deposition of the gold electrode on the extended graphite electrodes.

The transport properties of the NWs-MoTe₂-NWs field-effect transistor (FET) is shown in **Fig. R1** (as **Fig. 6** in the main text). An optical photo of the FET device is presented in **Fig. R1b** and the top- and side-views of the device schematic are shown in **Fig. R1c**. Source-drain I_{ds} - V_{ds} curve measurements [*i.e.*, drain-source current (I_{ds}) versus drain-source voltage (V_{ds})] at zero gate voltage

(inserted image in **Fig. R1d**) reveals a linear behavior with a resistance of approximately 0.9 M Ω , while a MoTe₂ device contacted directly with graphite electrodes (shown in **Fig. R2**) or vertical stacked NWs¹ show non-linear behavior. The corresponding gate-dependent I_{ds} - V_{gs} curve [*i.e.*, drain-source current (I_{ds}) versus gate voltage (V_{gs})] (**Fig. R1d**) exhibits intriguing ambipolar characteristics under gate voltage modulation. This behavior further underscores that the oriented NWs indeed improve the contact properties of the original 2H MoTe₂.¹ Furthermore, the carrier mobilities of the NW-MoTe₂-NW FETs can be extracted from the following Equation¹:

$$\mu = \left(\frac{L}{W}\right) \left(\frac{d}{\epsilon_0 \epsilon_r V_{ds}}\right) \left(\frac{dI_{ds}}{dV_{gs}}\right)$$

where a channel length (L) of 140.72 μm , width (W) of 19.63 μm , dielectric thickness (d) of 200 nm, permittivity (ϵ_r) of 3.4, and (dI_{ds}/dV_{gs}) of -1.26×10^{-8} A/V and 3.36344×10^{-9} A/V extracted from a linear fit of the transfer curve (I_{ds} - V_{gs} at a V_{ds} value of 1 V), are used in the calculation. The carrier mobility is calculated to be 5.99 $\text{cm}^2\text{V}^{-1}\text{s}^{-1}$ for holes, and 1.56 $\text{cm}^2\text{V}^{-1}\text{s}^{-1}$ for electrons. The drain-source current I_{ds} can be defined by the 2D thermionic emission equation²

$$I_{ds} = A_{2D}^* S T^{3/2} \exp \left[-\frac{q}{k_B T} \left(\phi_B - \frac{V_{ds}}{n} \right) \right],$$

where A_{2D}^* is the 2D equivalent Richardson constant, n is the ideality factor, and V_{ds} is the drain-source bias voltage. To determine the Schottky barrier height ϕ_B , it is common to use an Arrhenius plot, *i.e.*, $\ln(I_{ds}/T^{3/2})$ against $1000/T$ for various V_{ds} . Arrhenius graphs for the NWs-MoTe₂-NWs device at $V_{gs} = 0$ V from 200 to 300 K are shown in **Fig. R1e**. The slope S can be extracted as a function of V_{ds} , where $S = -\left(\frac{q}{1000k_B}\right) \left(\phi_B - \frac{V_{ds}}{n}\right)$. The Slope S can be plotted against V_{ds} , then after acquired the intercept $S_0 = -\frac{q\phi_B}{1000k_B}$, the Schottky barrier height can be determined. The Schottky barrier height in our case is **11.52 meV**, much smaller than the previously reported 1T'-2H-1T' results³⁻⁵ where the 1T' phase is used as contacts and comparable to the previously reported Pd/Mo₆Te₆/2H-MoTe₂ back-gated field-effect transistors ($q\phi_B = 8.7$ meV).¹ These newly added transport data showcase the novelty and potential applications of the oriented NWs.

Action 1-1: To address this issue, we have added related discussions on **Page 2**, **Page 4**, and **Pages 18-19** in the main text (highlighted by red) and **Fig. S18** on **Page 19** in the **Supporting Information**. The detailed sample preparation and transport measurements of the FET devices are added in the

Methods section (Pages 21-22 in the main text). We wrote:

“...Following the oriented NWs growth mechanism revealed by STEM, we fabricate NWs-MoTe₂-NWs field effect transistor (FET) devices through direct NWs patterning on MoTe₂, demonstrating the enhanced contact performance. Our work introduces possible ways to fabricate oriented NWs for interconnections in future flexible 2D nanoelectronics.” (Page 2 in the Abstract)

“...Furthermore, we fabricated NWs-MoTe₂-NWs FET devices through direct NWs patterning on MoTe₂. The transport results demonstrate an improvement in the device contact properties.” (on Page. 4)

Fig. R1. (Fig. 6 in the main text) **The transport results of the fabricated FET device by introducing NWs in the graphite electrode region.** Optical microscope images of the (a) fabricated graphite confined MoTe₂ and (b) NWs-MoTe₂-NWs field-effect transistor (FET) device, we introduce graphite confinement and fabricated NWs at both ends of the MoTe₂ through previous strategy. The gate graphite is isolated by h-BN. The OM image confirming the

conversion of NWs at the graphite edge is shown in the inserted image in **b**. **c**. Top- and cross-sectional view of the device schematic diagram. **d**. transfer curve ($I_{ds}-V_{gs}$) at $V_{ds}=1V$, with $I_{ds}-V_{ds}$ characteristics of the device at room temperature with $V_{gs} = 0$ inserted, showing linear behavior. **e**. Arrhenius plot $\ln(I_{ds}/T^{3/2})$ versus $1000/T$ at different values of V_{ds} (T vary from 300 K-200 K). **f**. extraction of $q\Phi_B$ via the intercept value, where each data point represents the slope obtained from the Arrhenius plot in (e) using a specific value of V_{ds} .

Fig. R2. The transport results of the graphite confined MoTe₂. Optical microscope images of the (a) fabricated graphite confined MoTe₂ and (b) fabricated device on silicon wafer. **c**. cross-sectional schematic diagram of the device. **d**. I-V curve measured from the device.

Comment 1-2. *The characterization of the 1D Mo₆Te₆ nanowires and 2D MoTe₂ is incomplete. In order to confirm the formation of materials, optical and crystallographic evidences have to be provided. Although the authors provided STEM and EDS data to characterize the 1D Mo₆Te₆ nanowires and 2D MoTe₂, I do not see the optical and crystallographic data such as Raman spectra, SAED patterns, and XRD patterns.*

Reply 1-2: In response to this suggestion, additional experiments have been conducted to characterize the 1D Mo_6Te_6 NWs and 2D MoTe_2 . The results of the selected area electron diffraction (SAED) are presented in **Fig. R3** (**Fig. S5** in the revised Supplementary Information). **Fig. R3a** displays TEM images highlighting the interface between 2H MoTe_2 and Mo_6Te_6 NWs after oriented conversion. The SAED patterns for both the NWs and MoTe_2 are depicted in **Figs. R3b-c**, agreeing with the corresponding simulations. The Raman spectra for 2H MoTe_2 and Mo_6Te_6 NWs are also illustrated in **Figs. R4** (**Fig. S6** in the revised Supplementary Information). Both the SAED and Raman data unambiguously confirm the successful conversion of 2H MoTe_2 to oriented 1D Mo_6Te_6 NWs. Notably, the Raman spectra data for the NWs presented in **Fig. R4b** reveal intriguing findings. In addition to the previously reported Raman peaks at 155 cm^{-1} and 245 cm^{-1} (highlighted by blue arrows),⁶ our oriented NWs exhibit two additional peaks at 90 cm^{-1} and 120 cm^{-1} (highlighted by red arrows). These extra peaks are believed to arise from the highly oriented nature of the sample. However, we are not able to provide the XRD data due to the limitation of the sample size ($30\mu\text{m}\times 30\mu\text{m}$).

Action 1-2: To address this issue, we have added related discussions on **Page 5** in the main text (highlighted by red) and data are added to **Figs. S5-6** on **Pages 6-7** of the revised **Supporting Information**.

Fig. R3. (**Fig. S5** in the revised Supplementary Information). **Selected area electron diffraction (SAED) results for 2H MoTe_2 and Mo_6Te_6 nanowires (NWs).** a. TEM image highlighting the interface between 2H MoTe_2 and Mo_6Te_6 NWs. b-c. SAED patterns corresponding to NWs and MoTe_2 with simulated SAED pattern inserted.

Fig. R4. (Fig. S6 in the revised Supplementary Information). **Raman spectra for 2H MoTe₂ and Mo₆Te₆ NWs.** **a.** OM image showcases the sample used for acquiring Raman spectra. **b-c.** Raman spectra for Mo₆Te₆ NWs and MoTe₂, two peaks marked by purple arrows are newly found in our oriented NW.

Comment 1-3. *They claimed that the origin of the phase transition is the formation of Te vacancies induced by Joule-heating. If the Te atoms were released from MoTe₂, there should be particles or clusters of Te atoms. Were you able to observe any of these?*

Reply 1-3: We are not able to observe any particles or clusters of the escaped Te atoms during the experiments. We suspect that the released Te atoms are quickly vaporized once they escape from the confinement region, whereby no particles or clusters are formed. In the experiments, the heterostructure was placed in the column of the microscope, which is in ultra-high vacuum of 10⁻⁵ pascal. Given that the melting point of Te element in ambient pressure is 450 °C, it becomes much lower in vacuum. Either electrical biasing or confined heating cause intense release of Te atoms from MoTe₂ and, when these Te atoms enter the gaps (NWs region), they are more likely to be compelled to migrate along the gap and rapidly vaporize from the heterostructure inside the microscope column.

Action 1-3: We have added related discussion on **Page 13** in the main text (highlighted by red) to illustrate that we do not see Te particles or clusters. We wrote:

“...We also notice that in either electrical biasing or confined heating experiments, no particles or clusters of the escaped Te atoms are observed. We infer that, when the escaped Te atoms enter the gaps (NWs region), they are more likely to migrate along the gap and rapidly vaporize from the heterostructure inside the ultra-high vacuum of the microscope.” (on **Page 13**)

Comment 1-4. *In this paper, the authors claimed that 2D 2H MoTe₂ is directly converted to 1D Mo₆Te₆. However, there is a paper (Nano Lett. 2018, 18, 2, 675–681) that reports that 1T' MoTe₂ could be generated in addition to 1D Mo₆Te₆ during the transition from 2H MoTe₂ to 1D Mo₆Te₆ induced by the formation of Te vacancies. In addition, both experiments and calculations have reported that the phase transitions from 2H MoTe₂ to 1T' MoTe₂ occur as the concentration of Te vacancies increases, (Nat. Commun., 2014, 5, 4214; Science, 2015, 349, 625-628; Adv. Mater. 2017, 29, 1605461). Is there a possibility that 2H MoTe₂ can be converted to 1T' MoTe₂ in the authors' work?*

Reply 1-4: In our experiments, we do not observe the phase transition from 2H MoTe₂ into 1T' MoTe₂ or any other intermediate phases. The phase transition in our experiments is clean and sharp. To address this issue, we carefully examined the references raised in this question. Although Te vacancies can drive different phase transformations in MoTe₂, the reason why we do not obtain the 1T' phase may be due to the different device schematic.

In the MoTe₂ system, the relatively small energy differences between phases open up opportunities for precise phase engineering. The literature mentioned in the comment achieved or predicted the phase transition to 1T' through the use of: (1) elevating the substrate temperature to approximately 450 °C during molecular beam epitaxy (MBE) synthesis,⁷ (2) laser irradiation⁸, (3) controlling the Te flux conditions during chemical vapor deposition (CVD) synthesis,⁹ and (4) applying tension along the 2H phase *b* axis (armchair).¹⁰ In sharp contrast, in our experiment, we employed vertical electric gating and graphite confinement to modulate the phase transition from the 2H phase of MoTe₂ to Mo₆Te₆ NWs. As illustrated in **Figs. 2e-f**, we observed a remarkably sharp and clean conversion with no discernible intermediate states. The intense generation of Te vacancies (compared with the references above) in our setup might be responsible for the absence of the 1T' phase, which needs further investigation but it is beyond the scope of this work.

Action 1-4: We added the discussion with newly cited references in the main text on **Page 13** (highlighted by red):

“It is also notable that unlike prior work that obtained transformations of MoTe₂ to other phases by vacancy generation,^{8,12,16,17,24-28} we only observe the direct transformation to oriented NWs, most likely because of the unique heterostructure we use to confine the material.” (on **Page 13**)

The newly cited references are listed below:

(12) Yu, Y.; Wang, G.; Tan, Y.; Wu, N.; Zhang, X.-A.; Qin, S. Phase-Controlled Growth of One-Dimensional Mo₆Te₆ Nanowires and Two-Dimensional MoTe₂ Ultrathin Films Heterostructures. *Nano Lett.* **2018**, 18 (2), 675-681.

(17) Cho, S.; Kim, S.; Kim, J. H.; Zhao, J.; Seok, J.; Keum, D. H.; Baik, J.; Choe, D.-H.; Chang, K. J.; Suenaga, K.; Kim, S. W.; Lee, Y. H.; Yang, H. Phase patterning for ohmic homojunction contact in MoTe₂. *Science* **2015**, 349 (6248), 625-628.

(27) Yoo, Y.; DeGregorio, Z. P.; Su, Y.; Koester, S. J.; Johns, J. E. In-Plane 2H-1T' MoTe₂ Homojunctions Synthesized by Flux-Controlled Phase Engineering. *Adv. Mater.* **2017**, 29 (16), 1605461.

(28) Duerloo, K.-A. N.; Li, Y.; Reed, E. J. Structural phase transitions in two-dimensional Mo- and W-dichalcogenide monolayers. *Nat. Commun.* **2014**, 5 (1), 4214.

Comment 1-5. *How thick is the graphite layers? Are there any possibility that the results will vary depending on the thickness of graphite layers?*

Reply 1-5: In the electrical biasing experiment, the thickness of the top- and bottom-layer graphite layer is 5.7 and 6.0 nm, and the MoTe₂ layer in the middle is 15.8 nm. We have repeated the same experiments over ten times with the same heterostructure configuration using different thickness in graphite and MoTe₂ layer, ranging from 5.6 to 17 nm. The oriented growth behavior of NWs remains consistent across varying thicknesses of graphite and MoTe₂ layers, as long as the MoTe₂ is graphite-confined on both sides. Thus, we concluded that there is no thickness-dependent effect in the

transition.

Action 1-5: We provided thickness evidence of the graphite and MoTe₂ layers of two typical *in-situ* devices, shown in **Fig. R5** (**Fig. S8** in the revised Supplementary Information). We have added these results on **Page 6** of the revised **Manuscript** and **Fig. S8** on **Page. 9** of the revised **Supporting Information**.

Fig. R5. (**Fig. S8** in the revised Supplementary Information) **The thickness of the graphite and MoTe₂ layers in the electric biasing experiment. a-b.** Cross-sectional high-angle annular dark field (HAADF) STEM images illustrating the converted NWs in two electric biasing experiments. The graphite layer thickness varies from 5.65 nm and 5.92 nm for the first biasing experiment and 17 nm for the second one.

Comment 1-6. *In Figure 3, low-magnification STEM images and some low-quality high-magnification STEM images are not enough to confirm the formation of 1D Mo₆Te₆ nanowires from 2D MoTe₂. The authors should provide additional data.*

Reply & Action 1-6: We have updated a revised **Fig. 3** in our revised manuscript to provide zoom-in atomic resolution images to confirm the formation of 1D Mo₆Te₆ in all of the experiments. The updated figure is shown in **Fig. R6**. Together with the additional data and discussions in **comment 1-2** and **comment 1-4**, the transition regions can be unambiguously determined as 1D Mo₆Te₆. To

further address this issue, supporting analysis has been included to elucidate the formation of 1D Mo_6Te_6 NWs from 2D MoTe_2 in global heating with confinement (**Fig. R7**), electrical biasing with confinement (**Fig. R8**), and heating without graphite confinement (**Fig. R9**), which are all incorporated as **Figs. S12-14** in the revised **Supporting Information**.

Fig. R6. (**Fig. 3** in the main text) **Different structural changes in controlled in-situ experiments.** Low- (**a.(i)**, **b.(i)**, and **c.(i)**) and high-magnification (**a.(ii)**, **b.(ii)**, and **c.(ii)**) HAADF-STEM images showcase the conversion of Mo_6Te_6 NWs from 2H MoTe_2 . **a.(i-ii)**. Phase overlap can be observed in heating with confinement experiment, with the FFT pattern inserted in **a.(ii)** (the FFT patterns highlighted by red dash line and yellow circle representing the 2H MoTe_2 and Mo_6Te_6 NW). **b.(i-ii)**. The interface structure in electrical biasing is clean and sharp. The inserted images in **b.(i)** and **b.(ii)** showcase the cross-sectional HAADF-STEM images of the gaps structure in NWs region, and the FFT pattern corresponding to **b.(ii)**. In heating without confinement experiments, NWs grow along the three zigzag edges [**a.(i)** and the inserted image]. Continued heating finally leads to the conversion of disordered NWs together with random size clusters as shown in **c.(ii)**.

Fig. R7. (Fig. S13 in the revised Supplementary Information) **The interface structure captured in the *in-situ* heating experiment.** **a-b.** High-resolution HAADF-STEM images offering a detailed view of the phase-overlapping at the MoTe_2 - Mo_6Te_6 NW interfaces. Corresponding fast Fourier transform (FFT) images are shown in the upper right. **c(i-ii).** Inverted FFT analysis images using the MoTe_2 and Mo_6Te_6 patterns in **b**, conclusively confirming the observed phase-overlapping.

Biasing with confinement

Fig. R8. (Fig. S12 in the revised Supplementary Information) **The interface structure captured in the *in-situ* electric biasing experiment.** Atomic-resolution HAADF-STEM image showcasing the (a) top-view and (b) cross-sectional sharp interface, with the corresponding FFT pattern shown in the upper right in a.

Heating without confinement

Fig. R9. (Fig. S14 in the revised Supplementary Information) **Interface structure captured in *in-situ* heating experiment without graphite confinement.** **a.** low-magnification STEM image shows the converted snow-flake-like Mo_6Te_6 NWs. **b.** atomic-resolution STEM image taken at the growth frontier of NW bundles (yellow square highlighted in **a**). **(c)** and **(d)** are taken at the nucleation point highlighted by a red square in **a**. Random size Mo clusters and disordered NWs can be found in **b-d**.

Reviewer #2 (Remarks to the Author):

Report for "Direct observation of controlled growth of oriented metal-chalcogenide nanowires " by Q. Yang et al. The authors report on growth of highly oriented Mo₆Te₆ nanowires. The conversion of Mo₆Te₆ NWs from MoTe₂ was thoroughly studied by STEM and first-principles calculations, which reveals that Te vacancy formation is the key. The characterization and explanation of the fabrication and properties of Mo₆Te₆ NWs were made in a convincing way. However, I do not recommend publication of this manuscript by its novelty and impact issues:

Reply 2-0: We sincerely appreciate your valuable suggestions, which have significantly contributed to the enhancement of our manuscript. In the beginning, we would like to humbly emphasize the novelty and difference of our work compared with others as follows:

1. In comparison to previously reported synthesis strategies for Mo₆Te₆ NWs based on thermal treatment, our work successfully achieved **direct metal-NW patterning with a well-defined orientation and a sharp interface** with MoTe₂ through on-device phase engineering without any pre-treatment.
2. Through the application of an electric field *via* graphite layers, we demonstrate the feasibility of introducing vertical electrical bias while concurrently recording the structural evolution of the sample in a STEM at the atomic scale using state-of-the-art MEMS-based *in-situ* technology, which have not been realized before.
3. For prototype device, the Schottky barrier height ($q\Phi_B$) we obtained in our newly fabricated NWs-MoTe₂-NWs heterostructure FET device is 11.52 meV, nearly half compared with previous reported 1T'-2H-1T' results and competitive to the reported Pd/Mo₆Te₆/2H-MoTe₂ back-gated field-effect transistor ($q\Phi_B = 8.7$ meV). This feature further demonstrates the novelty and potential applications of the oriented NWs.

Our findings indicate that it is indeed possible to regulate the growth direction of the NWs and provide an atomic scale understanding of the oriented growth. We believe that our work opens new avenues for the controlled manipulation and observation of nanomaterials, offering valuable insights for future nanodevice applications.

Comment 2-1. *The impact and motivation of this work are vague. The authors claimed that this kind of TMC-NWs would serve as conductive interconnections and decrease the Schottky barrier height in FETs. However, interconnect works in semiconductor industry and nanodevice engineering communities are not actively using these materials.*

Reply 2-1: We fully acknowledge your perspective that TMC-NWs have not yet been widely utilized in the current semiconductor industry. However, we contend that TMC-NWs could potentially serve as ultrasmall conductive interconnections in future nanoelectronics devices for the following reasons:

1. Previous research indicates that TMC-NWs exhibit tunable metallic characteristics.¹¹⁻¹⁴ In particular, Lin et al.'s publication in *Nature Nanotechnology* demonstrates the intrinsic metallic transition of the NWs from the semiconducting TMD monolayer. It has also been reported that introducing NWs to fabricate Pd/Mo₆Te₆/2H-MoTe₂ back-gated field-effect transistors led to high mobility of 1139 cm² V⁻¹ s⁻¹ with a low Schottky barrier height ($q\Phi_B$) of 8.7 meV.¹
2. TMC-NWs and TMDs share homogeneous compositions, implying that TMC-NWs can be patterned directly on TMD samples without the need for additional metal deposition or sample growth to serve as the conductive interconnections. Potentially, this not only simplifies the device fabrication process but also provides a new solution for enhancing device integration.

Therefore, TMC-NWs are promising candidates for ultrasmall interconnections in integrated circuits made by 2D materials, which have already been proposed by others previously, though still in a premature form for realistic applications. To clarify the efforts in exploring these NWs in the current fundamental research, we revised the sentence in the **Introduction** on **Page. 3** (highlighted by red) in the main text to emphasize the potential use of NWs in future micro-nano devices:

“TMC-NWs also show great structure stability and flexibility under extreme conditions such as electron beam irradiation.⁵ In addition, when assembling TMC-NWs with transition-metal dichalcogenides (TMDs) in the fabricated field-effect transistors (FETs), the Schottky barrier height between the TMDs and the metal electrode can be effectively lowered.⁷ Moreover, as TMC-NWs and TMDs share homogeneous compositions, TMC-NWs can be converted directly on TMDs samples

without the need for additional metal deposition to serve as the conductive interconnections.^{8,9} These unique features of TMC-NWs position them as promising conductive interconnects for future nanodevices.” (on **Page 3**)

Comment 2-2. *As for the Schottky barrier height study, there have been lots of studies using phase engineering (based on 1T-2H transition in MoTe₂), which is more versatile than this one; laser irradiation, strain, intercalation, and other methods enable this. To overcome this issue, the authors should provide more solid and quantitative comparison data with their Mo₆Te₆ and other MoTe₂ studies.*

Reply & Action 2-2: We agreed that the key idea in our work, compared with others, is the use of phase engineering of MoTe₂ to improve the contact performance. To further support the advancement of our study, we have collected previously reported studies (similar device schematic) on Mo₆Te₆ and other MoTe₂ studies. The data is presented in the table below. Compared to the reported work in the table, our study apparently stands out for the following two points:

1. We achieved direct on-device phase engineering by introducing graphite confinement in our MoTe₂ device.
2. Among all direct phase patterning methods with spatial control, the Schottky barrier height (Φ_B) in our device using oriented NWs as contact is 11.52 meV, nearly half compared with previous reported 1T'-2H-1T' results³⁻⁵, and competitive to the Pd/Mo₆Te₆/2H-MoTe₂ back-gated field-effect transistor ($q\Phi_B = 8.7$ meV), which nevertheless do not have any spatial control.¹ The detailed transport results of our device can be found in our reply to **Reviewer #1**, the **Reply 1-1**.

We have added related discussion on **Page 18** in the main text (highlighted by red) and comparison data are added as **Table S1** on **Page 20** of the revised **Supporting Information**.

References	Phase transition	Carrier Motilities (cm ² V ⁻¹ s ⁻¹)	Schottky barrier height (meV)	Direct phase patterning control
Adv. Funct. Mater. 2022, 32 (41), 2205299.	1T'/2H/1T'	5.6	37	Yes
Nat. Nanotechnol. 2017, 12 (11), 1064-1070	1T'/2H/1T'	16.2	25	No
ACS Nano 2019, 13 (7), 8035-8046	1T'/2H/1T'	7~8	30±10	No
Nano Lett. 2019, 19 (10), 6845-6852	1T'/2H/1T'	-	23	Yes
Science 2015, 349 (6248), 625-628.	2H-1T'	50	~10 (for 1T'), ~200 (for 2H)	Yes
Adv. Mater. 2017, 29 (16), 1605461.	2H-1T'	-	-	No
Nano Lett. 2018, 18 (2), 675-681.	Mo ₆ Te ₆ Nanowires	-	-	No
Adv. Mater. 2017, 29 (18), 1606264.	2H→Mo ₆ Te ₆ Nanowires	-	-	No
ACS Nano 2019, 13 (1), 642-648.	Mo ₆ Te ₆ Nanowires	1139	8.7	No
Our work	2H→Mo₆Te₆ Nanowires	5.99	11.52	Yes

Table R1. Comparisons of the performance of our NWs/2H/NWs FETs with other results reported in the literature.^{1,3-5,8,9,15-17}

Comment 2-3. *Many groups have reported the synthesis of similar TMDC nanowires (some of them are cited in this manuscript). And it is hard to see the advantages or novelty of Mo₆Te₆ NWs over them. Below are some details:*

(1) One paper (NanoLett.2020, 20, 8866) suggests W₆Te₆ nanowires show unique features such as semiconductor to metal transition, Luttinger liquid behaviors, which are based on its 1D confined geometry. This would be a good reason to use the nanowire. Are there similar benefits (e.g., fabricating 1D-2D-1D heterostructures) with Mo₆Te₆?

Reply & Action 2-3: As already mentioned before, the best benefit of fabricating oriented 1D NW to construct a 1D-2D-1D heterostructure is to greatly lower the contact Schottky barrier at the contact interface, while achieving spatial control of such on-device phase engineering. This has not been achieved in any previous research.

To clarify, we have added the transport data and discussion on **Pages 18-19** in the main text (highlighted by red) while the detailed sample preparation and transport data measurement of the FET device are added in the **Methods** section (**Pages 21-22** in the main text).

Comment 2-4. *The disordered Mo_6Te_6 NWs (Fig. 3c) show different transport? Some unique features like transport should be shown in this work for high impact journals.*

Reply & Action 2-4: In response to this concern, we have fabricated devices based on disordered NWs for comparison using open heating, as shown in **Fig. R10**. The I-V curves measured from the disordered NWs also show the same linear behavior. However, it is notable that the whole device becomes metallic as shown by the R - T curve in **Fig. R10d**, in sharp contrast to the gate-tunable bipolar semiconducting feature in the oriented NWs device shown in **Fig. R1**. Apparently, this is because NWs are randomly nucleated and disorderly generated inside the MoTe_2 flake if not confining their growth in the same orientation, resulting in short-circuiting the device. This further unequivocally showcase the unique feature of the oriented NWs for directional electrodes in constructing nanoelectronics devices.

Fig. R10. **The transport results of the converted disordered Mo_6Te_6 NWs.** Optical microscope images of the (a) fabricated disordered NWs on MoTe_2 , b. fabricated device on silicon wafer (insert OM image shows the zoom-in feature of NWs). c. I-V curve measured from the device at room temperature. d. the device resistance curve with changing the temperature from 300K to 8K, showing the metallic property.

Reviewer #3 (Remarks to the Author):

The manuscript describes the oriented growth of Mo₆Te₆ nanowires (NWs) from a graphite-encapsulated MoTe₂ heterostructure driven by local Joule-heating induced by electrical bias, using in-situ Scanning Transmission Electron Microscopy (STEM), Density Functional Theory (DFT) calculations, and Molecular Dynamics (MD) simulations. It shows a detailed analysis of nanowire growth. However, the submitted manuscript does not provide any new significant research findings compared to what's been published by other researchers. In particular, the exhibited results are very similar to the one by Ref. 14 (Zhu et al., Adv. Mater. 2017, 29, 1606264). The only difference is whether the main activation energy for nanowire phase transformation is provided by direct heating or joule heating induced by electrical biasing. And yet, no estimate is given for the joule heating temperature induced by electrical bias. Other findings are almost identical, such as the metallic nature of bundled nanowires vs. the small gap semiconductor nature of an isolated nanowire; a sharp interface between Mo₆Te₆ nanowire and 2D MoTe₂; intercalated chalcogen atoms along the gap between nanowires in transition to a stable energy minimized configuration. The highly oriented growth of nanowires presented in the submitted manuscript is noteworthy, though.

Reply 3-0: We greatly appreciate the constructive suggestions, which have been instrumental in enhancing our manuscript. Compared to **Ref. 14** (Zhu et al., Adv. Mater. 2017, 29, 1606264), the novelty of our work is reflected in the following aspects:

1. We achieved direct phase patterning of oriented Mo₆Te₆ NWs by introducing graphite confined layers. In **Ref. 14**, applying vacuum annealing can only achieve the fabrication of disordered NWs with random distribution, which is not suitable for device construction (see also our reply to **Reviewer #2 in Reply 2-4**); whereas with the confinement effect applied by graphite layers, we achieved precise oriented NW creation in the graphite confined region. Achieving precise patterning of NWs is essential for the effective utilization of TMC-NWs in future micro-nano devices.
2. Using similar strategy of the *in-situ* device for TEM application, we fabricated NWs-MoTe₂-NWs FET device and measured the Schottky barrier and carrier mobility. The Schottky barrier height ($q\Phi_B$) we obtained is 11.52 meV, nearly half compared with previously reported 1T'-2H-1T'

device results and competitive to the reported Pd/Mo₆Te₆/2H-MoTe₂ back-gated field-effect transistor ($q\Phi_B = 8.7$ meV). This feature demonstrates the unique ability of oriented NWs in improving contacts and reducing Schottky barriers, while the Ref. 14 does not provide relevant data on device performance. The detailed transport results can be found in the **Reply 1-1**.

3. Through theoretical and DFT calculations, we conduct an in-depth study regarding the growth mechanism of highly oriented NWs. Compared with disordered phase transitions shown in **Ref. 14**, by constructing graphene-confined heterojunctions, we show that it is possible to regulate the migration path of the escaped Te atoms.

In summary, compared to **Ref. 14**, we believe that the on-device phase engineering method presented in our work showcases the unique properties of highly oriented NWs. **Reviewer #2** also raised similar concerns, which we have addressed by summarizing the novelty of our work point-by-point. For details, please see our reply to **Reviewer #2** in **Reply 2-0**.

Comment 3-1. *The following couple of general comments are provided for the authors to consider: The electrical in-situ STEM experiment is based on the heterostructure attached to MEMS chips. The vertical field is achieved by the top and bottom graphite layers in the heterostructure are connected to the electrodes. In Fig.2 (a), and Fig. S1, the bottom layer graphite (red marked) seems to connect with negative electrode as expected, and the metal wire linked to the positive electrode as well. However, the resistance vs time (Fig.1 c) does not show the short feature. Any efforts help with insulation and prevent short? Could the author provide more details on the MEMS circuit and the current path?*

Reply 3-1: Thank you for pointing out this mistake. We have corrected the position of the graphite layers shown in **Fig. 2a** and Supplementary **Fig. S1**. The corrected image is presented in **Fig. R11a**. It is essential to note that the top graphite-confined layers (marked as red dashed lines) were not connecting to both biasing electrodes, since it is being blocked by the MoTe₂ layer (marked as black dashed lines) in the middle, as illustrated in the schematic in **Fig. R11b**. According to our design, the current flow initially passes through the top-layer graphite, then traverses the MoTe₂ layer from top to bottom as highlighted by green dash line in **Figs. R11a-b**. After the conversion of Mo₆Te₆ NWs,

the current flow follows a path from the top-layer graphite, crosses the converted Mo_6Te_6 region from top to bottom, and ultimately passes through the bottom layer graphite. OM image of the MEMS chip is shown in **Fig. R11c**. Two electrodes highlighted by black arrows are used for applying bias voltage.

Action 3-1: We have corrected the image information in **Fig. 2a**, the discussion on **Pages. 8-9** in the main text (highlighted by red), and **Fig. S1** on **Page. 2** of the revised **Supporting Information**.

Fig. R11. (Fig. S1 in the revised Supplementary Information) **Details related to the MEMS chip and sample preparation.** **a.** An optical image showcases the heterostructure positioned on the MEMS chip. The sections delineated by red and green dashed lines signify the upper and lower graphite layers, respectively. The specimen demarcated by blue dashed lines denotes the few-layers MoTe_2 . The region shaded in blue designates the defined heterostructure area where MoTe_2 is covered by graphite on both sides. The device cross-sectional schematic is shown in **b**. **c.** Optical microscope images presenting the electrode positions (black arrows highlighted). The current path is highlighted by green dash line in **a-b**.

Comment 3-2. *The HAADF image exhibits fringe contrast (Fig.1 (b.i)), how to interpret the contrast? Is there any ripple feature on the heterostructure that may affect the strain & stress of materials? A micrometer scale cross-section image or AFM data on the heterostructure would help to better understand the graphite-sandwiched MoTe_2 structure.*

Reply & Action 3-2: **Fig. R12** showcase the surface rippling of the heterostructure in large scale. The fringe contrast in **Fig. 1(b.i)** come from the surface undulation on the free-standing graphite/ MoTe_2 /graphite heterostructure, which could be due to the following reasons:

1. The sample cleavage process (scotch tape cleavage) and transfer process, which includes picking up and dropping down the samples several cycles, introduce certain cracks and wrinkles for thick layers.
2. The diameter of the free-standing holes is around 10 μm . The lack of a supporting substrate induces certain surface rippling after making the 2D heterostructure free-standing.

We have added related discussions on **Page 5** in the main text (highlighted by red) and data are added to **Fig. S4** on **Page 5** of the revised **Supporting Information**. We wrote:

“...Although slight deflection of certain stripes is observed due to surface undulations (see Fig. S4)...”
(on **Page 5**)

Fig. R12. (Fig. S4 in the revised Supplementary Information) **Surface rippling of the heterostructure in the free-standing hole.** a-b. Cross-sectional HAADF-STEM images present the sample geometry. The interface between the 2H-MoTe₂ and Mo₆Te₆ NWs is shown in the upper-right in a.

Comment 3-3. *The authors attribute the formation of the overlapping interface is caused by the temperature gradient. What is the thickness of the MoTe₂ layer? Could the author give an estimation of the temperature difference between top and bottom? The temperature difference might not be significant at the nanoscale dimension.*

Reply 3-3: Thank you for pointing out this important issue. First, the thickness of the MoTe₂ layers used in the *in-situ* electric biasing experiment is 20 nm thick. It is nearly impossible to directly measure the chip's exact temperature due to the design of the chips and holder (utilizing a 4-pins holder with two for heating and two for electric biasing), not to mention the temperature gradient. Therefore, we fully agree that, at the nanoscale, temperature gradients may not be that significant. The preferential creation of vertical overlapping interfaces in the heating experiment may be caused by various factors, predominantly resulting from different heating treatments, as electrical biasing generates localized Joule heating near the interface, while direct heating can be considered as more global.

Action 3-3: We deleted the expression of “temperature gradient” and revised the sentence on **Page 13** in the main text to reflect this change:

“It seems that vertical conversion of NWs is more thorough in the electrical biasing case, as sharp MoTe₂/NWs connected interface is always present (Fig. 2e and Fig. 3b), while an overlapping interface is persistently seen in the case of global heating (Fig. 3a), presumably due to localized vs. global heating condition.” (on **Page 13**)

Comment 3-4. *The author compared the result between in-site anneal and in-site electric STEM experiment. And conduct the electrical biasing case may engender more defect that result into an intense transformation and form a sharp interface. However, the in-situ experiment on heterostructure is complex that the authors need to provide more evidence to convince reader.*

What the is thickness of the Graphite layer and MoTe₂ layer used in both experiments?

Reply & Action 3-4: In biasing experiments, as shown in **Fig. R13**, the thickness of the top- and bottom-layer graphite layer is 5.7 and 5.9 nm, and the MoTe₂ layer in the middle is 15.8 nm. In heating with confinement experiment, according to the flake color observed through OM images, the thickness is around 15 nm for the top and bottom graphite layers and 20 nm for MoTe₂.

We have added these data in **Figs. S8 (Page. 9** in the revised **Supporting Information**).

Fig. R13. (Fig. S8 in the revised Supplementary Information) The thickness of the graphite and MoTe₂ layers in the electric biasing experiment. a-b. Cross-sectional high-angle annular dark field (HAADF) STEM images illustrating the converted NWs in two electric biasing experiments. The graphite layer thickness varies from 5.65 nm and 5.92 nm for the first basing experiment and 15-16 nm for the second one.

Comment 3-5. *Is the interface structure between MoTe₂ and NWs bias or temperature related? Or ramping related?*

Reply 3-5: In response to your inquiry, the interface structure between MoTe₂ and NWs is not related to the variations in bias and temperature in all of our *in-situ* experiments. We have conducted electrical bias *in-situ* experiments in the range 0-12 V, and heating experiments in the range 600-750 °C. When the bias or heating temperature reach the threshold value, the NWs undergo sharp conversion from 2H MoTe₂ to NWs. The interface structure remains almost similar even in different ramping rates of the bias voltage as well as the temperature. As pointed out in our reply to **comment 3-3**, the difference in the interface structure may be primarily attributed to the differences in NW growth rates induced by interfacial Joule localized heating (local Joule heating) versus global heating effect (direct heating from the chip).

Comment 3-6. *Any direct comparison with anneal experiment, that exhibit electrical experiment is*

“more intense?” Like serial images showing phase transformation area vs time?

Reply & Action 3-6: In response to this concern, we calculated the phase transformation area vs time for four *in-situ* movies. The results are shown in **Fig. R14**. Below are several points that we want to clarify from our perspective:

1. Changing the bias voltage and heating temperature (above threshold value) results in altering the growth speed of NWs, as illustrated by the slopes in **Fig. R14e** and the data presented in **Fig. R14f**. Higher temperature causes faster growth rate than the electrical-bias experiments. Therefore, the term “more intense” does not refer to the NW growth rate (or growth speed).
2. In fact, electrical bias generated a much sharper interface structure compared to the heating, as illustrated by the atomic images shown in **Fig. 2e** and **Fig. 3b**. As pointed out in our reply to **comment 3-3** and **3-5**, the electrical bias method always generates sharper interfaces. Therefore, in the main text, “more intense” refers to the formation of a sharper and cleaner interface. (in the biasing case)

To avoid the above ambiguity, we have modified our descriptions for the comparison between electrical bias and heating on the terminology of “intense” during the conversion of NWs. We refrain from using the term “intense” to prevent readers from mistakenly associating it with a higher growth rate. In the revised manuscript, we added these data as **Figs. S15** and rewrote the sentence on **Page 11** and **Page 13** in the main text (highlighted by red) as follows:

“...The difference in the interface structure in these two experiments may be caused by various factors, predominantly resulting from different heating treatments. In biasing experiments, as current flows through the converted NWs region and induces local joule heating, the electrical biasing may engender more localized defect generation at the interface, while direct heating can be considered as more global. We also find that changing the bias voltage and heating temperature (above threshold value) will only result in altering the growth speed of NWs, as illustrated in Fig. S15, higher temperature will have faster growth rate than the electrical bias experiment.” (on **Page. 11**)

“...It seems that vertical conversion of NWs is more thorough in the electrical biasing case, as sharp

MoTe₂/NWs connected interface is always present (Fig. 2e and Fig. 3b), while an overlapping interface is persistently seen in the case of global heating (Fig. 3a), presumably due to localized vs. global heating condition....” (on Page. 13)

Fig. R14. (Fig. S15 in the revised Supplementary Information) Phase transformation area vs time in the biasing and heating experiments. HAADF-STEM presenting the phase transformation area vs time in (a-b) electrical

biasing, and (c-d) heating experiments. The phase transformation area vs time are shown in e. f. The statistical data regarding the transformation and growth rate. The scale for curve d (blue curve in e) is μm^2 . (see **Supplementary Movies 1, 3-5**)

Comment 3-7. *The author shall clearly indicate the advance or the uniqueness of the proposed electrical induced phase transformation strategies.*

The concept of electrical induced phase transformation and confinement effect on MoTe_2 material has been widely discussed in published works (Nature, 2017, 550(7677): 487-491, Ref 18 in this manuscript) (Nature materials, 2019, 18(1): 55-61) (Advanced Functional Materials, 2021, 31(51): 2107376) (Nano letters, 2023, 23(2): 677-684). I would propose to discuss these studies and highlight the innovation and significance of the present work.

Reply 3-7: We thank the constructive suggestion on the related reference. In the MoTe_2 system, the relatively small energy differences between phases open up opportunities for precise phase engineering. The literature mentioned in the reviewer's comments achieved the phase transition through: (1) liquid ion gating induced reversible phase transition between 2H and 1T'.¹⁸ (2) electric-field induced phase transition from 2H towards H_d phase facilitated by gate electrode.¹⁹ (3) h-BN encapsulated annealing induced phase transition ($2\text{H} \rightarrow \text{T}_d$).²⁰ (4) reversible phase engineering between 2H and T_d phase using laser irradiation ($2\text{H} \rightarrow \text{T}_d$) and thermal annealing ($\text{T}_d \rightarrow 2\text{H}$).²¹

As already pointed out in our reply to **comment 1-4** from the **Review #1**, we emphasize that in our experiments, the phase transition towards oriented Mo_6Te_6 NWs is clean and sharp without any intermediate states. We would like to humbly emphasize the novelty and difference of our work compared with other published works (confinement effect and phase engineering) as follows:

1. In comparison to previously reported synthesis strategies, through on-device phase engineering, our approach enables the precise control of phases in the fabrication of oriented Mo_6Te_6 NWs by introducing graphite confinement to MoTe_2 without any pre-treatment such as laser irradiation or heating. Besides, our method also enables the **direct metal-phase patterning** on devices, providing a strategic avenue for the potential utilization of TMC-NWs in future nanodevices.
2. Through the application of an electric field *via* graphite layers, we demonstrate the feasibility of introducing vertical electric bias while concurrently recording the atom-resolved structural

evolution of the sample in a STEM, which have not been achieved in any previous work.

3. Apart from conducting theoretical and DFT calculations regarding the phase transition mechanism, we have now also fabricated NWs-MoTe₂-NWs FET devices and measured the Schottky barrier and carrier mobility outside the electron microscope using similar strategy. The Schottky barrier height ($q\Phi_B$) we obtained in our newly fabricated graphite-MoTe₂-graphite heterostructure FET device is 11.52 meV, nearly half compared with previous reported 1T'-2H-1T' results,³⁻⁵ demonstrating the introduction of NWs can enhance the device contact.

We believe that our work contributes to the direct phase patterning through applying graphite confinement and provide alternative *in-situ* characterization strategy by introducing vertical electrical gating.

Action 3-7: We have added these related references in the **Introduction** and discussed them for comparison on **Page 4** and **Page 13** (highlighted by red). We have also revised the content in the **Abstract (Page 2)** and **Discussions (Page 20)** sections of the main text (highlighted by red). In the main text, we wrote:

“...In this work, we report the realization of successive transition from 2D molybdenum ditelluride (MoTe₂) to highly oriented Mo₆Te₆ NWs driven by *in-situ* electrical bias voltage through confining graphite layers in a scanning transmission electron microscope (STEM).” (on **Page 4**)

“It is also notable that unlike prior work that obtained transformations of MoTe₂ to other phases by vacancy generation,^{8,12,16,17,24-28} we only observe the direct transformation to oriented NWs, most likely because of the unique heterostructure we use to confine the material.” (on **Page 13**)

“...Following the oriented NWs growth mechanism revealed by STEM, we fabricate NWs-MoTe₂-NWs field effect transistor (FET) devices through direct NWs patterning on MoTe₂, demonstrating the enhanced contact performance. Our work introduces possible ways to fabricate oriented NWs for interconnections in future flexible 2D nanoelectronics.” (on **Page 2** in the **Abstract**)

“...Furthermore, through direct oriented NWs patterning on MoTe₂ inspired by our STEM results, NWs-MoTe₂-NWs FET devices were fabricated to show that the introduction of NWs can significantly enhance the device contact performance. Our work introduces the potential that NWs can serve as interconnections in future flexible 2D nanoelectronics.” (on **Page 20**)

The newly cited references are listed below:

16. Wang Y, *et al.* Structural phase transition in monolayer MoTe₂ driven by electrostatic doping. *Nature* **550**, 487-491 (2017).
24. Zhang F, *et al.* Electric-field induced structural transition in vertical MoTe₂- and Mo_{1-x}W_xTe₂-based resistive memories. *Nat. Mater.* **18**, 55-61 (2019).
25. Ryu H, *et al.* Anomalous Dimensionality-Driven Phase Transition of MoTe₂ in Van der Waals Heterostructure. *Adv. Funct. Mater.* **31**, 2107376 (2021).
26. Lee C-H, *et al.* In Situ Imaging of an Anisotropic Layer-by-Layer Phase Transition in Few-Layer MoTe₂. *Nano Lett.* **23**, 677-684 (2023).

References

1. Lee, R.S., Kim, D., Pawar, S.A., Kim, T., Shin, J.C., Kang, S.-W. van der Waals Epitaxy of High-Mobility Polymorphic Structure of Mo₆Te₆ Nanoplates/MoTe₂ Atomic Layers with Low Schottky Barrier Height. *ACS Nano*, **13**, 642-648 (2019)
2. Anwar, A., Nabet, B., Culp, J., Castro, F. Effects of electron confinement on thermionic emission current in a modulation doped heterostructure. *Journal of Applied Physics*, **85**, 2663-2666 (1999)
3. Ma, R., et al. MoTe₂ Lateral Homo Junction Field-Effect Transistors Fabricated using Flux-Controlled Phase Engineering. *ACS Nano*, **13**, 8035-8046 (2019)
4. Xu, X., et al. Scaling-up Atomically Thin Coplanar Semiconductor–Metal Circuitry via Phase Engineered Chemical Assembly. *Nano Lett.*, **19**, 6845-6852 (2019)
5. Sung, J.H., et al. Coplanar semiconductor–metal circuitry defined on few-layer MoTe₂ via polymorphic heteroepitaxy. *Nat. Nanotechnol.*, **12**, 1064-1070 (2017)
6. Kim, H., Johns, J.E., Yoo, Y. Mixed-Dimensional In-Plane Heterostructures from 1D Mo₆Te₆ and 2D MoTe₂ Synthesized by Te-Flux-Controlled Chemical Vapor Deposition. *Small*, **16**, 2002849 (2020)
7. Yu, Y., Wang, G., Tan, Y., Wu, N., Zhang, X.-A., Qin, S. Phase-Controlled Growth of One-Dimensional Mo₆Te₆ Nanowires and Two-Dimensional MoTe₂ Ultrathin Films Heterostructures. *Nano Lett.*, **18**, 675-681 (2018)
8. Cho, S., et al. Phase patterning for ohmic homo junction contact in MoTe₂. *Science*, **349**, 625-628 (2015)
9. Yoo, Y., DeGregorio, Z.P., Su, Y., Koester, S.J., Johns, J.E. In-Plane 2H-1T' MoTe₂ Homo Junctions Synthesized by Flux-Controlled Phase Engineering. *Adv. Mater.*, **29**, 1605461 (2017)
10. Duerloo, K.-A.N., Li, Y., Reed, E.J. Structural phase transitions in two-dimensional Mo- and W-dichalcogenide monolayers. *Nat. Commun.*, **5**, 4214 (2014)
11. Vilfan, I. Mo₆S₆ nanowires: structural, mechanical and electronic properties. *Eur. Phys. J. B*, **51**, 277-284 (2006)
12. Çakır, D., Durgun, E., Gülseren, O., Ciraci, S. First principles study of electronic and

- mechanical properties of molybdenum selenide type nanowires. *Phys. Rev. B*, **74**, 235433 (2006)
13. Kibsgaard, J., et al. Atomic-Scale Structure of Mo₆S₆ Nanowires. *Nano Lett.*, **8**, 3928-3931 (2008)
 14. Lin, J., et al. Flexible metallic nanowires with self-adaptive contacts to semiconducting transition-metal dichalcogenide monolayers. *Nat. Nanotechnol.*, **9**, 436-442 (2014)
 15. Zhang, S., et al. Field Effect Transistor Sensors Based on In-Plane 1T'/2H/1T' MoTe₂ Heterophases with Superior Sensitivity and Output Signals. *Adv. Funct. Mater.*, **32**, 2205299 (2022)
 16. Tan, Y., et al. Controllable 2H-to-1T' phase transition in few-layer MoTe₂. *Nanoscale*, **10**, 19964-19971 (2018)
 17. Zhu, H., et al. New Mo₆Te₆ Sub-Nanometer-Diameter Nanowire Phase from 2H-MoTe₂. *Adv. Mater.*, **29**, 1606264 (2017)
 18. Wang, Y., et al. Structural phase transition in monolayer MoTe₂ driven by electrostatic doping. *Nature*, **550**, 487-491 (2017)
 19. Zhang, F., et al. Electric-field induced structural transition in vertical MoTe₂- and Mo_{1-x}W_xTe₂-based resistive memories. *Nat. Mater.*, **18**, 55-61 (2019)
 20. Ryu, H., et al. Anomalous Dimensionality-Driven Phase Transition of MoTe₂ in Van der Waals Heterostructure. *Adv. Funct. Mater.*, **31**, 2107376 (2021)
 21. Lee, C.-H., et al. In Situ Imaging of an Anisotropic Layer-by-Layer Phase Transition in Few-Layer MoTe₂. *Nano Lett.*, **23**, 677-684 (2023)

REVIEWER COMMENTS

Reviewer #1 (Remarks to the Author):

The authors have addressed all my concerns. I fully support publication of this manuscript in Nature Communications.

Reviewer #2 (Remarks to the Author):

Report for "Direct observation of controlled growth of oriented metal-chalcogenide nanowires " by Q. Yang et al.

The authors replied to my comments to a certain extent, but the impact of this work is still not enough for publication in Nature Communications.

They repeatedly mention "good stability" and "lowering the Schottky barrier" by growing Mo₆Te₆ nanowires and structuring them into the device. However, similar things have been reported in multiple papers (as summarized in Table R1) and I cannot find other novel (e.g., nanowire growth or 1D transport) or promising (e.g., scalability) aspects of this work. This work is worth being reported but in a more specialized journal.

Reviewer #3 (Remarks to the Author):

The response letter has good information that addresses some of the reviewers' comments. However, most of the new information is not shown in the manuscript or the SI. Also, the contact resistance between the Mo₆Te₆ and MoTe₂ is low, as demonstrated by other previous publications. A more critical contact-related issue is the contact resistance between the electrode and the Mo₆Te₆. I would strongly suggest you redraft the manuscript to focus more on the contact issues and relevant device characteristics. The current version is mostly related to the graphite-guided conversion of MoTe₂ to Mo₆Te₆, which is incremental for high-impact papers like Nature Comm.

Reviewer #4 (Remarks to the Author):

I co-reviewed this manuscript with one of the reviewers who provided the listed reports.

This is part of the Nature Communications initiative to facilitate training in peer review and to provide appropriate recognition for Early Career Researchers who co-review manuscripts.

Response to reviewers' comments (NCOMMS-23-41661B)

Title: Confined patterning of oriented metal-chalcogenide nanowires and its atomic in-situ growth mechanism for superior contact properties

Authors: Qishuo Yang, Yun-Peng Wang, Xiao-Lei Shi, XingXing Li, Erding Zhao, Zhi-Gang Chen, Jin Zou, Kai Leng, Yongqing Cai, Liang Zhu,* Sokrates T. Pantelides, Junhao Lin*

We sincerely thank the reviewers for the constructive comments on our revised manuscript. We have addressed all the comments point-by-point and revised the manuscript accordingly. In this Response Document, comments from the referees are in **regular black** typeface, our responses are in regular **blue** typeface. All major changes have been highlighted in **red** in the main text and supplementary information (SI). A list of the main changes in the manuscript and SI is provided below:

1. We fully redrafted the manuscript to focus more on the contact issues and relevant device characteristics and reorganized Figs. 1-2 and 6. We demonstrate the incorporation of oriented NWs in devices and then elucidate the growth mechanism through *in-situ* STEM characterization and DFT calculations. We also changed the title of the manuscript to reflect the purpose of the study.
2. We measured the contact resistance between Mo₆Te₆ NWs and four different combinations of common metal electrodes. We find that Au electrodes-Mo₆Te₆ NWs exhibit low contact resistance (43.7 Ω·μm), which is the lowest recorded among previous findings using the 1T' phase as contact.
3. We provided more data in comparing our results with previous work, demonstrating that NWs are promising candidates as contacts in future flexible 2D nanoelectronics with low Schottky barrier and contact resistance.

We sincerely express our gratitude once again to the reviewers for your valuable time, and we believe that all your concerns have been addressed in this response.

Reviewer #1 (Remarks to the Author):

The authors have addressed all my concerns. I fully support publication of this manuscript in Nature Communications.

Reply 1-0: We sincerely appreciate the recognition from the reviewer. Thanks again for the previous comments that have improved the manuscript.

Reviewer #2 (Remarks to the Author):

Report for "Direct observation of controlled growth of oriented metal-chalcogenide nanowires " by Q. Yang et al.

The authors replied to my comments to a certain extent, but the impact of this work is still not enough for publication in Nature Communications.

They repeatedly mention “good stability” and “lowering the Schottky barrier” by growing Mo_6Te_6 nanowires and structuring them into the device. However, similar things have been reported in multiple papers (as summarized in Table R1) and I cannot find other novel (e.g., nanowire growth or 1D transport) or promising (e.g., scalability) aspects of this work. This work is worth being reported but in a more specialized journal.

Reply 2-0: We appreciate the recognition by the reviewer of our experimental results and the relevant technical issues, which confirms the reliability of our transport data and *in-situ* characterization results. In this revised version, we have fully characterized the contact properties of NWs at the metal-NWs interface by conducting more transport measurements. We believe that a comparison with previous strategies using 1T'-2H contacts suggests that our oriented Mo_6Te_6 NWs hold promise for superior contacts in TMD-based devices as follows:

1. In comparison to previously reported strategies based on thermal treatments, we achieved **direct metal-NW patterning** with a **well-defined orientation** through introducing graphite confined layers. Controlling the orientation from random to unidirectional is the prerequisite for realistic applications of these NWs in devices. Therefore, our method lays the foundation for constructing **lateral contacts** of NWs in 2H MoTe_2 FET devices.
2. To highlight the novelty of Mo_6Te_6 NWs transport properties, we provided extra data regarding the **contact resistance** between Mo_6Te_6 NWs and different combination of common metal electrodes (see Fig. R1 below). In our case, comparing with the contact resistance between 1T'/NWs and metal electrodes, we obtain the lowest contact resistance among previous findings (**$43.7 \Omega \cdot \mu\text{m}$**).¹⁻³ The Schottky barrier is **11.5 meV**, which is nearly halved compared with the

previously reported 1T' contact, and also comparable to the vertically stacked Pd/Mo₆Te₆ NWs/2H-MoTe₂ back-gated FET which has a Schottky barrier of 8.7 meV, but much higher contact resistance of $R_c=28.7\text{ M}\Omega$ (see Tabel S1 for detailed comparison).⁴ The small contact resistance at the metal-NW interface and relatively low Schottky barrier in the NW-2H contact FET device demonstrate the potential of NWs that can serve as superior contacts in future nanodevices

3. Besides the exploration of NW transport properties, we conducted in-depth investigation of the physical mechanisms regarding the oriented growth of Mo₆Te₆ NWs by integrating *in-situ* atomic-resolution STEM data with theoretical calculation analysis, which is essential for further extending the impact of our work. We confirm the significant role played by the graphite confined layers in this phase transition process and local Joule heating induced by vertical electrical biasing. In summary, our work not only presents novel insights for improving device contacts but also deepens the understanding of the confined-growth process in nanostructures down to the atomic scale.

Fig. R1. Transport results of the Mo_6Te_6 NWs and fabricated (NW- MoTe_2 -NW) FET device. a. Transfer curve ($I_{ds}-V_{ds}$) of Mo_6Te_6 NWs device, showing linear behaviour. The inserted $R-T$ curve demonstrates the metallic nature of bulk Mo_6Te_6 NWs. **b.** Contact resistance between Mo_6Te_6 NWs and Au+Ti, Au+Cr, Cr+Ti, and Pd+Ti metal electrodes. The Ti and Cr metal are deposited as adhesion layer with thickness around 6 nm. **c.** Schematic of our NWs- MoTe_2 -NWs field-effect transistor (FET) device. **d.** Optical-microscope images of the fabricated graphite confined MoTe_2 heterostructure, and **(e)** the fabricated FET device. We introduce graphite confinement and fabricated oriented NWs at both ends of the MoTe_2 . An optical-microscope image confirming the conversion of oriented NWs at the graphite edge is shown in the inserted image in **e**. **f.** Transfer curve ($I_{ds}-V_{gs}$) at $V_{ds}=1\text{V}$, with $I_{ds}-V_{ds}$ characteristics of the device at room temperature with $V_{gs} = 0$ inserted, showing linear behaviour. **g.** Arrhenius plot $\ln(I_{ds}/T^{3/2})$ versus $1000/T$ at different values of V_{ds} (T vary from 300 K-200 K). **h.** Extraction of $q\Phi_B$

via the intercept value, where each data point represents the slope obtained from the Arrhenius plot in **g** using a specific value of V_{ds} . **i**. Comparison of contact resistance between 1T'/NWs-metal electrodes and Schottky barrier height of the 1T'/NWs-2H contact FET of this work with previously reported results (ref 8, 9, 16, 18, 21, 36, 38-42 in the Manuscript). The devices with 1T'-2H MoTe₂ contact are highlighted by yellow ellipse, and NW-2H contact devices are highlighted by black arrows. Detailed comparison is provided in Table S1 (Table R1 shown below).

References	Phase transition	Carrier Motilities (cm ² V ⁻¹ s ⁻¹)	Schottky barrier height (meV)	Contact Resistance (Ω.μm)	Direct phase patterning control
Adv. Funct. Mater. 2022, 32 (41), 2205299. ³⁸	1T'/2H/1T'	5.6	37	7.1x10 ³	Yes
Nat. Nanotechnol. 2017, 12 (11), 1064-1070. ³⁹	1T'/2H/1T'	16.2	25	1.4x10 ⁴	No
ACS Nano 2019, 13 (7), 8035-8046. ⁹	1T'/2H/1T'	7~8	30±10	235 (1T') 7.8x10 ⁶ (2H)	No
Nano Lett. 2019, 19 (10), 6845-6852. ⁴⁰	1T'/2H/1T'	-	23	1.1x10 ³	Yes
Science 2015, 349 (6248), 625-628. ⁸	2H-1T'	50	~10 (for 1T'), ~200 (for 2H)	~1x10 ³ (2H contact) ~100 (1T' contact)	Yes
Adv. Mater. 2017, 29 (16), 1605461. ³⁶	2H-1T'	-	-	-	No
ACS Appl. Nano Mater. 2020, 3, 10, 10411–10417. ⁴¹	2H-1T'	15	-	3.64 x10 ⁴ (1T'/2H) 8x10 ³ (1T')	Yes
Science , 372, 195-200 (2021). ⁴²	1T'-2H	45	6.15 (Vg=-55V)	1.6x10 ³ (1T')	No
Nano Lett. 2018, 18 (2), 675-681. ¹⁸	Mo ₅ Te ₆ Nanowires	-	-	-	No
Adv. Mater. 2017, 29 (18), 1606264. ²¹	2H→Mo ₅ Te ₆ Nanowires	-	-	-	No
ACS Nano 2019, 13 (1), 642-648. ¹⁶	Mo ₅ Te ₆ Nanowires	1139	8.7	2.85x10 ⁷	No
Our work	2H→Mo₅Te₆ Nanowires	5.99	11.52	43.73	Yes

Table R1. Comparisons of the contact resistance of 1T'/NWs phase and the performance of our NWs/2H/NWs FETs with other results reported in the literature. (ref 8, 9, 16, 18, 21, 36, 38-42 in the Manuscript)

Action 2-0: We have added related discussions on **Pages 16-28** in the main text (highlighted by red) and data are incorporated as **Fig. 5** in the **Manuscript**. The detailed comparison data are shown in **Table S1**.

Reviewer #3 (Remarks to the Author):

The response letter has good information that addresses some of the reviewers' comments. However, most of the new information is not shown in the manuscript or the SI. Also, the contact resistance between the Mo_6Te_6 and MoTe_2 is low, as demonstrated by other previous publications. A more critical contact-related issue is the contact resistance between the electrode and the Mo_6Te_6 . I would strongly suggest you redraft the manuscript to focus more on the contact issues and relevant device characteristics. The current version is mostly related to the graphite-guided conversion of MoTe_2 to Mo_6Te_6 , which is incremental for high-impact papers like Nature Comm.

Reply 3-0: We sincerely thank the reviewer for the comments after carefully reading our revised manuscript and response letter. We fully agree with the reviewer that we need to highlight the good electrical properties of oriented NWs. Following this thought, we fully redrafted the manuscript and rewrote the **Abstract** and **Introduction** to focus more on the contact issue of current 2D devices, while the metallic NWs hold the promise to substantially improve the contact. We also changed the order of the figures and the title of the manuscript to reflect the purpose of the study. In this revised version, we have fully characterized the contact properties of NWs at the metal-NWs interface by conducting more transport measurements on the contact resistance among different metal combinations deposited electrodes. In addition, to better help readers understand the major problem of NWs for serving as contact in 2D TMDs devices and our original ideas to solve this problem, we provide extra schematics of our unique growth method as shown in Fig. R2 (Figs. 1 in the **Manuscript**). We also summarized the superior electrical properties of NWs in Fig. R3 (Fig. 5 in the **Manuscript**).

For the convenience of the reviewer, we summarized the measurements of the contact resistance between Mo_6Te_6 NWs and different combinations of common metal electrodes here. We find that Au electrodes- Mo_6Te_6 NWs exhibit low contact resistance (minimal **43.7 $\Omega\cdot\mu\text{m}$**), which is the lowest recorded among previous findings using 1T'-metal electrode contact. The corresponding data can be seen in Fig. R3. Our data demonstrate that introducing NWs to fabricate NWs-2H lateral contact device can effectively lower the Schottky barrier as well as the contact resistance at the same time.

We reorganized the DFT calculations and MD simulation results (Figs. 4-5 in the previous version

of the manuscript); the new figure is shown below as Fig. R4 (**Fig. 6 in the revised Manuscript**). We have also carefully revised the manuscript and SI to incorporate all the data from the previous response letter, such as the MEMS circuit, comparison of the growth rate, extra transport results of NWs, and so on.

In summary, by introducing graphite encapsulation layers, we have achieved for the first-time control over the growth orientation of one-dimensional NWs. Additionally, we combined atomic-resolution *in-situ* STEM data with theoretical calculations to meticulously investigate the growth mechanism of the NWs. Furthermore, through direct NW patterning on MoTe₂, we fabricated a series of devices, demonstrating the superior contact properties of these NWs. Therefore, we believe that our work will appeal to a broad audience in the fields of 2D materials and nanodevices, which fits the scope of *Nature Communications*.

Fig. R2. Control growth of oriented Mo₆Te₆ NWs through introducing graphite confined layers. a(i-iii). Schematics of the conversion of disordered NWs through applying thermal annealing. **b(i-iii).** Schematics of the conversion of oriented NWs in the graphite-encapsulated 2H-MoTe₂. Potential Te migration pathways in a graphite-confined MoTe₂ flake are highlighted by orange arrows in b(ii), including 1) penetrating graphite layers, 2) along NWs, and 3) along the gaps between NW bundles.

Fig. R3. Transport results of Mo_6Te_6 NWs and fabricated (NW- MoTe_2 -NW) FET device. a. Transfer curve ($I_{ds}-V_{ds}$) of Mo_6Te_6 NWs device, showing linear behaviour. The inserted $R-T$ curve demonstrates the metallic nature of bulk Mo_6Te_6 NWs. **b.** Contact resistance between Mo_6Te_6 NWs and Au+Ti, Au+Cr, Cr+Ti, and Pd+Ti metal electrodes. The Ti and Cr metal are deposited as adhesion layer with thickness around 6 nm. **c.** Schematic of our NWs- MoTe_2 -NWs field-effect transistor (FET) device. **d.** Optical-microscope images of the fabricated graphite confined MoTe_2 heterostructure, and (e) the fabricated FET device. We introduce graphite confinement and fabricated oriented NWs at both ends of the MoTe_2 . An optical-microscope image confirming the conversion of oriented NWs at the graphite edge is shown in the inserted image in e. **f.** Transfer curve ($I_{ds}-V_{gs}$) at $V_{ds}=1\text{V}$, with $I_{ds}-V_{ds}$ characteristics of the device at room temperature with $V_{gs} = 0$ inserted, showing linear behaviour. **g.** Arrhenius plot $\ln(I_{ds}/T^{3/2})$ versus $1000/T$ at different values of V_{ds} (T vary from 300 K-200 K). **h.** Extraction of $q\Phi_B$

via the intercept value, where each data point represents the slope obtained from the Arrhenius plot in **g** using a specific value of V_{ds} . **i.** Comparison of contact resistance between 1T'/NWs-metal electrodes and Schottky barrier height of the 1T'/NWs-2H contact FET of this work with the previous reported results (ref 8, 9, 16, 18, 21, 36, 38-42 in the Manuscript). The devices with 1T'-2H MoTe₂ contact are highlighted in the yellow ellipse, while NW-2H contact devices are highlighted by black arrows. Detailed comparison is provided in Table S1.

Fig. R4. Growth mechanism of oriented NWs by controlled *in-situ* experiments. Low- (a, and c) and high-magnification (b, and d) HAADF-STEM images showcasing the conversion of Mo₆Te₆ NWs from 2H MoTe₂. **a-b** 2H phase region overlap can be observed in heating with confinement, with the FFT pattern inserted in **b** (the FFT

patterns highlighted by red dash line and yellow circle representing the 2H MoTe₂ and Mo₆Te₆ NWs). **c-d.** The interface structure in electrical biasing is clean and sharp. The inserted images in **c** showcase the cross-sectional HAADF-STEM images of the gap structure in the NW region. The inserted image in **d** is the FFT pattern corresponding to the STEM image in **d**. The FFT pattern regarding to 2H MoTe₂ and Mo₆Te₆ NWs are highlighted by red dash lines and yellow circles. **e.** High-resolution STEM image showing triangular- and line-shaped defects formed in the MoTe₂ region (pointed by yellow arrows) near the growth frontier during electrical biasing. **f-j.** Molecular dynamics simulation of the Mo₆Te₆ NW formation in a MoTe₂ lattice with Te deficiency. **h.** DFT-calculated energy barrier for a Te atom migrating along NWs (blue line highlighted) and across three close-packed Mo₆Te₆ NWs (red line highlighted). The migration energy barriers are calculated by changing the x-axis coordinate of the Te atoms. (x=abscissa position of the Te atom). The corresponding schematics are shown in **i-j**.

Action 3-0: We added new **Fig. 1** to illustrate the confined strategy as serial schematics on **Page 6** in the **Manuscript**. The transport results including the updated contact resistance are summarized and presented in **Fig. 5** on **Pages 16-18** in the **Manuscript**. We update the MEMS circuit in **Fig. 2a** on **Page. 9** in the **Manuscript**. We update the thickness of the graphite and MoTe₂ layers in the heating experiments on **Page 12** in the **Manuscript**. We reorganized the DFT-calculation and MD-simulation results and data are shown in **Fig. 4** on **Page 15** in the **Manuscript**. Abstract and Introduction of the manuscript is rewritten, and title is also changed. More discussions on the contact performance are included.

Reference:

1. Ma, R., et al. MoTe₂ Lateral Homojunction Field-Effect Transistors Fabricated using Flux-Controlled Phase Engineering. *ACS Nano*, **13**, 8035-8046 (2019)
2. Xu, X., et al. Scaling-up Atomically Thin Coplanar Semiconductor–Metal Circuitry via Phase Engineered Chemical Assembly. *Nano Letters*, **19**, 6845-6852 (2019)
3. Sung, J.H., et al. Coplanar semiconductor–metal circuitry defined on few-layer MoTe₂ via polymorphic heteroepitaxy. *Nature Nanotechnology*, **12**, 1064-1070 (2017)
4. Lee, R.S., Kim, D., Pawar, S.A., Kim, T., Shin, J.C., Kang, S.-W. van der Waals Epitaxy of High-Mobility Polymorphic Structure of Mo₆Te₆ Nanoplates/MoTe₂ Atomic Layers with Low Schottky Barrier Height. *ACS Nano*, **13**, 642-648 (2019)

REVIEWERS' COMMENTS

Reviewer #2 (Remarks to the Author):

The author redrafted the manuscript, focusing on the contact resistance issue. Indeed, optimizing the contact material with Mo₆Te₆ makes competitive device performance. Some remaining issues, such as scaling up the contact process and verifying the work function difference between Mo₆Te₆ and Cr, Ti, could be followed in future studies. So, now I recommend the publication of this manuscript in Nature Communications.

Reviewer #4 (Remarks to the Author):

The manuscript has met all necessary criteria and addressed previous concerns. I recommend it for publication in Nature Communications as it is.